# Conditioning Sparse Variational Gaussian Processes for Online Decision-making

**Wesley J. Maddox**
New York University
wjm363@nyu.edu

**Samuel Stanton**
New York University
ss13641@nyu.edu

**Andrew Gordon Wilson**
New York University
andrewgw@cims.nyu.edu

## Abstract

With a principled representation of uncertainty and closed form posterior updates, Gaussian processes (GPs) are a natural choice for online decision making. However, Gaussian processes typically require at least $\mathcal{O}(n^2)$ computations for $n$ training points, limiting their general applicability. Stochastic variational Gaussian processes (SVGPs) can provide scalable inference for a dataset of fixed size, but are difficult to efficiently condition on new data. We propose online variational conditioning (OVC), a procedure for efficiently conditioning SVGPs in an online setting that does not require re-training through the evidence lower bound with the addition of new data. OVC enables the pairing of SVGPs with advanced look-ahead acquisition functions for black-box optimization, even with non-Gaussian likelihoods. We show OVC provides compelling performance in a range of applications including active learning of malaria incidence, and reinforcement learning on MuJoCo simulated robotic control tasks.

## 1 Introduction

Intelligent systems should be able to quickly and efficiently adapt to new data, adjusting their prior beliefs in response to the most recent events. These characteristics are desirable whether the system in question is controlling the actuators of a robot, tuning the power output of a laser, or monitoring the changing preferences of users on an online platform. What these applications share in common is a constant stream of new information. In this paper, we are interested in efficient *conditioning*, meaning that we wish to efficiently update a posterior distribution after receiving new data.

The ability of Gaussian process (GP) regression models to condition on new data in closed form has made them a popular choice for Bayesian optimization (BO), active learning, and control [24]. All of these settings share similar characteristics: there is an "outer loop", where new data is acquired from the real world (e.g. an expensive simulator), interleaved with an "inner loop", which chooses where to collect data. In BO, for example, the "inner loop" is the optimization of an acquisition function evaluated using a surrogate model of the true objective. Simple acquisition functions, e.g. expected improvement (EI), consider only the current state of the surrogate, while more sophisticated acquisition functions "look ahead" to consider the effect of hypothetical observations on future surrogate states. One such acquisition function, batch knowledge gradient (qKG), defines the one-step Bayes-optimal data batch as the batch that maximizes the expected surrogate maximum *after* the batch has been acquired [2, 84]. Advanced acquisition functions like qKG require the surrogate to have both efficient posterior sampling and efficient conditioning on new data.

GP regression has two major limitations that have prevented its large scale deployment for online decision-making. First, the computational and memory consumption of exact GPs grows at least quadratically with the amount of data [25, 65], generally limiting their usage to BO problems with fewer than $1,000$ function evaluations [24, 2, 79]. Second, they are limited to applications that have continuous real-valued responses, enabling modelling with solely a Gaussian likelihood. Stochastic

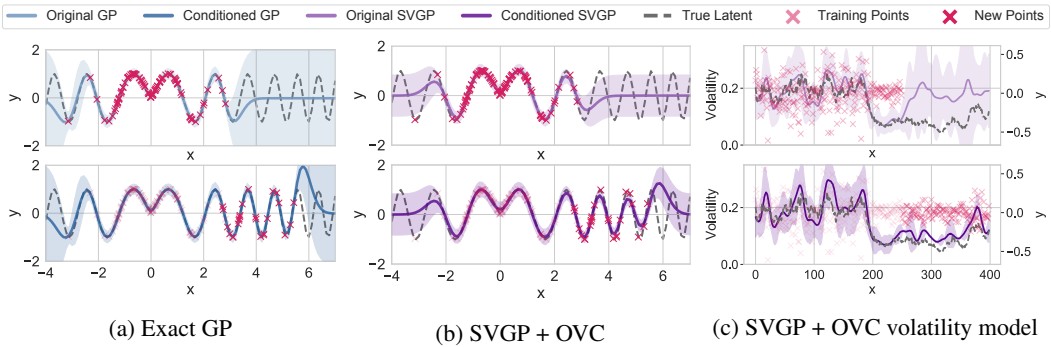

|   |   |   |
|---|---|---|
| (a) Exact GP | (b) SVGP + OVC | (c) SVGP + OVC volatility model |

Figure 1: An exact GP updates its predictive distribution after conditioning on new data points (**a,** moving from top row to bottom row). With OVC, we can condition SVGPs on both Gaussian responses (**b**) and non-Gaussian models (**c**) such as the Gaussian copula volatility model [82].

variational Gaussian processes (SVGPs) [33] have constant computational and memory footprints and are applicable to non-Gaussian likelihoods, but they sacrifice closed form expressions for updated posteriors on receiving new data. The SVGP posterior is *optimized* through the evidence lower bound (ELBO). In the online setting, training with the ELBO has two primary difficulties: the need to specify a fixed number of observations to properly scale the ELBO gradient [8] and the need to adjust the inducing points without looking at past data [3, 9]. Thus, we are presented with a choice between the simplicity and analytic tractability of exact GPs and the scalability and flexibility of SVGPs.

In this work, we develop **O**nline **V**ariational **C**onditioning (OVC) to allow SVGPs to be conditioned on-the-fly, as shown in Figure 1. In the top row of each subplot, we fit the data points shown in red, shifting to another batch of data points in the bottom row. We use an exact GP in Figure 1a, with exact conditioning shown in the bottom panel. The SVGP emulates the exact GP very well before seeing the new data and again after conditioning on the new data by using OVC (Figure 1b). In Figure 1c, we consider a non-Gaussian data model (a Gaussian copula volatility model [82]), where we cannot use exact GPs; the SVGP is still able to update its posterior over the latent volatility in response to new data without "forgetting" old observations. OVC is inspired by a new, simple rederivation of streaming sparse GPs (O-SGPR), originally proposed by Bui et al. [9]. OVC makes SVGPs truly compelling models for online decision-making, augmenting their existing strengths with efficient, closed-form conditioning on new data points. In short, our contributions are:

- The development of OVC, a novel method to condition SVGPs on new data without re-optimizing the variational posterior through an evidence lower bound.
- OVC provides both stable inducing point initialization for SVGPs while enabling the inducing points and variational parameters to update in response to the new data.
- Enabling the effective application of SVGPs, through OVC, in look-ahead acquisitions in BO for black-box optimization, controlling dynamical systems, and active learning.

Please see Appendix A for discussion of the limitations and broader impacts of our work. Our code is available at `https://github.com/wjmaddox/online_vargp`.

## 2  Preliminaries and Related Work

In this section, we first review exact inference with Gaussian processes before reviewing variational sparse Gaussian processes, introducing sparse Gaussian process regression (SGPR), sparse variational Gaussian processes (SVGP), and streaming sparse GPs (O-SGPR). Please see Appendix B for further background and Appendix B.1 specifically for further related work.

### 2.1  Gaussian Processes and Exact Inference

For a full introduction to GPs, see Rasmussen and Williams [65]. We begin by reviewing GP regression, with inputs $\mathbf{x} \in \mathcal{X}$ and responses $y \in \mathbb{R}$. GP regression assumes $y \sim \mathcal{N}(f, \sigma^2)$, where

$f \sim \mathcal{GP}(0, k_\theta(\mathbf{x}, \mathbf{x}'))$ is a GP characterized by the kernel function $k_\theta$ and corresponding hyperparameters $\theta$. Given $X_{\text{train}} = [\mathbf{x}_1, \ldots, \mathbf{x}_n]^\top$ and $X_{\text{test}} = [\mathbf{x}_1', \ldots, \mathbf{x}_k']^\top$, we denote the corresponding function values as $\mathbf{v} = [f(\mathbf{x}_1), \ldots, f(\mathbf{x}_n)]^\top$ and $\mathbf{w} = [f(\mathbf{x}_1'), \ldots, f(\mathbf{x}_k')]^\top$, respectively. Note that $\mathbf{v}$ and $\mathbf{w}$ are random variables with prior covariance $\text{Cov}[\mathbf{v}, \mathbf{w}] = K_{\mathbf{vw}} := k_\theta(X_{\text{train}}, X_{\text{test}})$. A well-known identity for multivariate conditional Gaussians allows us to compute the GP predictive posterior $p(\mathbf{w}|X_{\text{test}}, \mathcal{D}, \theta) = \mathcal{N}(\mu_{\mathbf{w}|\mathcal{D}}^*, \Sigma_{\mathbf{w}|\mathcal{D}}^*)$ as follows:

$$\mu_{\mathbf{w}|\mathcal{D}}^* = K_{\mathbf{wv}}(K_{\mathbf{vv}} + \sigma^2 I)^{-1}\mathbf{y}, \quad \Sigma_{\mathbf{w}|\mathcal{D}}^* = K_{\mathbf{ww}} - K_{\mathbf{wv}}(K_{\mathbf{vv}} + \Sigma_{\mathbf{y}})^{-1}K_{\mathbf{vw}}, \quad (1)$$

where $\mathcal{D} := (X_{\text{train}}, \mathbf{y})$ and $\Sigma_{\mathbf{y}} = \sigma^2 I$.

Naïvely, computations with $K_{\mathbf{vv}}$ cost $\mathcal{O}(n^2)$ space and $\mathcal{O}(n^3)$ operations. When repeatedly computing $p(\mathbf{w}|X_{\text{test}}, \mathcal{D}, \theta)$ at different sets of query points $X_{\text{test}}$, it is more efficient to cache the terms which depend only on the training data.[1] Specifically, we store $\mathbf{a} := (K_{\mathbf{vv}} + \Sigma_{\mathbf{y}})^{-1}\mathbf{y}$ (the predictive mean cache) and $RR^\top := (K_{\mathbf{vv}} + \Sigma_{\mathbf{y}})^{-1}$ (the predictive covariance cache), resulting in simplified forms for the predictive distribution, $\mu_{\mathbf{w}|\mathcal{D}}^* = K_{\mathbf{wv}}\mathbf{a}$ and $\Sigma_{\mathbf{w}|\mathcal{D}}^* = K_{\mathbf{ww}} - K_{\mathbf{wv}}RR^\top K_{\mathbf{vw}}$. Adding a new observation is equivalent to adding a single row and column to $K_{\mathbf{vv}}$ and an entry to $\mathbf{y}$, which enables efficient low-rank updates to the predictive caches [58, 25, 64, 39].[2]

## 2.2 Variationally Sparse Gaussian Processes

Variational sparse GPs reduce the computational burden of GP inference through sparse approximations of the kernel matrix. For further reference, see Matthews [50] and Van der Wilk [77]. These methods define a variational distribution $\phi$ over the inducing point *values* $\mathbf{u} = [f(\mathbf{z}_1), \ldots, f(\mathbf{z}_p)]^\top$, defined at inducing point *locations* $Z = [\mathbf{z}_1, \cdots, \mathbf{z}_p]^\top$, where $\mathbf{z}_i \in \mathcal{X}$. $\phi(\mathbf{u})$ is parameterized as a Gaussian with variational mean and covariance $\mathbf{m}_{\mathbf{u}}$ and $S_{\mathbf{u}}$. These methods assume the latent function values $f(\mathbf{x}), f(\mathbf{x}')$ are conditionally independent given $\mathbf{u}$ and $\mathbf{x}, \mathbf{x}' \notin Z$, so as to cheaply approximate the predictive posterior $p(\mathbf{w}|X_{\text{test}}, \mathcal{D}, \theta) \approx q(\mathbf{w}) = \mathcal{N}(\mu_{\mathbf{w}|\phi}^*, \Sigma_{\mathbf{w}|\phi}^*)$. Like exact GP regression, we can compute $q(\mathbf{w})$ in closed form (given $\mathbf{m}_{\mathbf{u}}, S_{\mathbf{u}}$),

$$\mu_{\mathbf{w}|\phi}^* := K_{\mathbf{wu}}K_{\mathbf{uu}}^{-1}\mathbf{m}_{\mathbf{u}}, \quad \Sigma_{\mathbf{w}|\phi}^* := K_{\mathbf{ww}} - K_{\mathbf{wu}}K_{\mathbf{uu}}^{-1}(K_{\mathbf{uu}} - S_{\mathbf{u}})K_{\mathbf{uu}}^{-1}K_{\mathbf{uw}}. \quad (2)$$

Similarly the predictive mean and covariance caches are given by $\mathbf{a} = K_{\mathbf{uu}}^{-1}\mathbf{m}_{\mathbf{u}}$ and $RR^\top = K_{\mathbf{uu}}^{-1}(K_{\mathbf{uu}} - S_{\mathbf{u}})K_{\mathbf{uu}}^{-1}$, reducing the complexity of inference from $\mathcal{O}(n^3)$ to $\mathcal{O}(np^2)$, which is a significant improvement if $p \ll n$.

There are two common approaches to finding optimal variational parameters $\mathbf{m}_{\mathbf{u}}$ and $S_{\mathbf{u}}$. In seminal work, Titsias [73] proposed sparse GP regression (SGPR), which optimizes $\mathbf{m}_{\mathbf{u}}$ and $S_{\mathbf{u}}$ in closed form, resulting in a "collapsed" evidence lower bound[3] (ELBO) that only depends on $\theta$ and $Z$. The computational cost of each gradient update to the remaining model parameters is still linear in $n$, and like exact GP regression, SGPR requires a Gaussian likelihood. Stochastic variational GPs (SVGPs) remedies both these limitations by using gradient-based optimization to learn $\mathbf{m}_{\mathbf{u}}$ and $S_{\mathbf{u}}$ alongside $Z$ and $\theta$ [33, 34]. The SVGP objective is an "uncollapsed" ELBO which decomposes additively across the training examples, allowing gradients to be estimated from minibatches of data, reducing the complexity of each gradient update to $\mathcal{O}(bp^2 + p^3)$, where $b$ is the minibatch size.

We emphasize the distinction between constant-time minibatch gradients, and constant-time conditioning. Given an SVGP already trained on some existing data, conditioning jointly on both the old and new data requires storing all the data and making multiple gradient updates to the variational parameters. As the size of the dataset grows, so does the number of gradient steps needed. In contrast by constant-time conditioning we mean a procedure that takes a posterior conditioned on existing data and produces a new posterior conditioned jointly on the old and new data with a fixed amount of compute and memory, regardless of the number of past observations.

One example of constant-time conditioning is found in Bui et al. [9], who proposed streaming sparse GPs (which we call online SGPR, or O-SGPR, to distinguish from sparse spectrum GPs [44]) for

---

[1]Cached expressions will be written in orange

[2]What we use as the SGPR covariance cache is slightly different from the implementation in the prediction strategy in GPyTorch, which stores $R = K_{\mathbf{vu}}K_{\mathbf{uu}}^{-1/2}$. However, they reduce to the same strategy.

[3]A lower bound of the true GP marginal log-likelihood

---

**Algorithm 1** Online Variational Conditioning (OVC)

---

**Input:** Data batch $(X_{\text{batch}}, \mathbf{y})$, SVGP with inducing points $Z'$ and $\phi(\mathbf{u}') = \mathcal{N}(\mathbf{m}_{\mathbf{u}'}, S_{\mathbf{u}'})$.

1. Compute $\mathbf{c}'$, $C'$ (Eq. 5).
2. Compute $\hat{\mathbf{y}} = K'_{\mathbf{u}'\mathbf{u}'} C'^{-1} \mathbf{c}'$ and $\Sigma_{\hat{\mathbf{y}}} = K'^{-1}_{\mathbf{u}'\mathbf{u}'} C' K'^{-1}_{\mathbf{u}'\mathbf{u}'}$ (Eq. 8).
3. Construct GP with $\mathcal{D} = ([X_{\text{batch}} \ Z'], [\mathbf{y} \ \hat{\mathbf{y}}])$ and $\Sigma = \text{blkdiag}(\Sigma_{\mathbf{y}}, \Sigma_{\hat{\mathbf{y}}})$.
4. Compute predictive mean and covariance caches, $a$ and $RR^{\top}$ as in Section 2.1.
5. Use caches to compute conditioned GP posterior on test points, $X_{\text{test}}$.

---

incremental learning. We extend their work, providing an alternative, simpler derivation of their model that highlights the connection with SGPR [73]. Furthermore, our perspective enables us to construct a principled approach to updating inducing point locations as new data arrives, that prevents the "forgetting" of old data induced by the resampling heuristic used by Bui et al. [9].

### 2.3 Bayesian Optimization and Monte Carlo Acquisitions

Bayesian optimization (BO) obtains $\mathbf{x}^* = \operatorname{argmin}_{\mathbf{x} \in \mathcal{X}} f(\mathbf{x})$ by constructing a probabilistic *surrogate model* of $f$, which in turn is used to evaluate an acquisition function. GPs are favored for BO due to their sample-efficiency and efficient posterior sampling that enables cheap, gradient based optimization of the acquisition function to propose new query points [24]. Many interesting acquisition functions look ahead into the future to see how the model will change if we query a specific point, a procedure known as "fantasization" [32, 84, 39]. Fantasization is done by drawing samples from the current surrogate posterior at some set of points and conditioning the surrogate on those samples. For example, the batch knowledge gradient [qKG, 84, 2] is given by

$$a(\mathbf{x}, \mathcal{D}) := \mathop{\mathbb{E}}_{f(\mathbf{x}) \sim p(\cdot|\mathcal{D})} \left( \max_{\mathbf{x}' \in \mathcal{X}} \mathop{\mathbb{E}}_{f(\mathbf{x}') \sim p(\cdot|\mathcal{D}_{+\mathbf{x}})} f(\mathbf{x}') \right) - \max_{\mathbf{x}' \in \mathcal{X}} \mathop{\mathbb{E}}_{f(\mathbf{x}') \sim p(\cdot|\mathcal{D})} f(\mathbf{x}'), \tag{3}$$

where $\mathcal{D}_{+\mathbf{x}} := \mathcal{D} \cup \{(\mathbf{x}, f(\mathbf{x})\}$. The inner expectation in the first term requires conditioning the surrogate model on posterior samples at $\mathbf{x}$, before optimizing through predictions of the conditioned surrogate model. The goal is to simulate the effect on the model if we had observed the batch of data.

**Use of sparse GPs in BO:** Sparse GPs have not seen wide adoption in the BO community, with only several preliminary studies that have mostly used basic acquisitions. Nickson et al. [55] and Krityakierne and Ginsbourger [43] used expected improvement (EI) with SGPR on several test problems, while McIntire et al. [51] proposed a sparse GP method using EI to tune free electron lasers [20]. Stanton et al. [70] proposed WISKI, an online implementation of a scalable kernel approach called SKI [83], for low-dimensional BO problems using batch upper confidence bound (qUCB) [2].

## 3 Methodology

We now briefly describe the key ideas behind OVC with the goal of devising an efficient and stable method for updating the variational parameters with respect to newly observed data. We begin by highlighting an alternative parameterization of SGPR that will prove useful. Then we describe the OVC update to the variational distribution from two equivalent points of view, namely the *projection view* and the *pseudo-data view*. The pseudo-data view is summarized in Algorithm 1. Next we address a critical detail for good performance, which is how the inducing point locations should be selected. We then demonstrate how OVC can be applied to compute updated posterior distributions, e.g. $p(f|\mathcal{D}_{+\mathbf{x}})$ in Eq. 3, and quantities of the posterior, during gradient-based acquisition function optimization in BO, with reference to how this can be performed practically in Section 4. Finally, we discuss how to apply OVC to models with non-Gaussian likelihoods.

### 3.1 Updating the Variational Posterior

We assume that we have trained a SVGP model (e.g. with the ELBO) on a fixed set of data and have already trained the inducing point locations and variational parameters, $\mathbf{m}_{\mathbf{u}}, S_{\mathbf{u}}$. Instead of the traditional $\mathbf{m}_{\mathbf{u}}, S_{\mathbf{u}}$ parameterization used by Titsias [73], Hensman et al. [33, 34], we focus for now on an alternative parameterization which was favored in early work on sparse GP inference

[67, 57]. The parameterization is also similar to those used in both dual space functional variational inference [41] and expectation propagation [10]. More recently, Panos et al. [61] used a similar parameterization in the context of large scale multi-label learning with SVGPs.

The SGPR predictive posterior $q(\mathbf{w})$ relies on two terms dependent on the training data,

$$\mathbf{c} = K_{\mathbf{uv}}\Sigma_{\mathbf{y}}^{-1}\mathbf{y}, \qquad C = K_{\mathbf{uv}}\Sigma_{\mathbf{y}}^{-1}K_{\mathbf{vu}}, \tag{4}$$

where $\Sigma_{\mathbf{y}}$ is the covariance of the likelihood $p(\mathbf{y}|\mathbf{f})$. The optimal $\mathbf{m_u}$, $S_{\mathbf{u}}$ are then given by

$$\mathbf{m_u} = K_{\mathbf{uu}}(K_{\mathbf{uu}} + C)^{-1}\mathbf{c}, \qquad S_{\mathbf{u}} = K_{\mathbf{uu}}(K_{\mathbf{uu}} + C)^{-1}K_{\mathbf{uu}}, \tag{5}$$

which can be substituted into Eq. (2) to obtain $q(\mathbf{w})$. Our first observation is that if $\Sigma_{\mathbf{y}}$ is block-diagonal, then $\mathbf{c}$ and $C$ are additive across blocks of observations. For some intuition, consider i.i.d. Gaussian noise (i.e. $\Sigma_{\mathbf{y}} = \sigma^2 I_n$), which implies

$$\mathbf{c}_i = \sum_j \sigma^{-2}y_j k_\theta(\mathbf{z}_i, \mathbf{x}_j) = \phi(\mathbf{z}_i)^\top \sum_j \sigma^{-2}y_j\phi(\mathbf{x}_j),$$

$$C_{ik} = \sum_j \sigma^{-2}k_\theta(\mathbf{z}_i, \mathbf{x}_j)k(\mathbf{x}_j, \mathbf{z}_k) = \phi(\mathbf{z}_i)^\top \sum_j \sigma^{-2}\phi(\mathbf{x}_j)\phi(\mathbf{x}_j)^\top \phi(\mathbf{z}_k),$$

where $\phi$ is the (potentially infinite-dimensional) feature map associated with $k_\theta$. Hence the entries of $\mathbf{c}$ and $C$ are both inner products between projected inducing points and weighted sums of features. For fixed inducing points, $Z$, and hyper-parameters $\theta$, we can use these updates to produce a streaming version of SGPR by exploiting the additive structure of $c$ and $C$. Furthermore, this streaming version of SGPR is exactly Gaussian conditioning for SGPR as we show in Appendix C.1. We can also allow the inducing points and hyper-parameters to vary, which we address next.[4]

**The projection view:** We assume we have $\mathbf{c}' = K'_{\mathbf{u'v'}}\Sigma_{\mathbf{y'}}^{-1}\mathbf{y}'$ and $C' = K'_{\mathbf{u'v'}}\Sigma_{\mathbf{y'}}^{-1}K'_{\mathbf{v'u'}}$, computed with inducing point locations $Z'$ from data $(X'_{\text{batch}}, \mathbf{y}')$ with kernel hyperparameters $\theta'$ (using shorthand $k_{\theta'} = K'$).[5] After obtaining the next parameters $Z$ and $\theta$ (perhaps from gradient based optimization of the ELBO), we observe new data $(X_{\text{batch}}, \mathbf{y})$ and would like to continue with inference. One challenge is translating $\mathbf{c}', C'$ (whose elements are inner products of the old features) to the new feature space associated with $\theta$. To resolve this challenge, we construct a representative set of responses, $\hat{\mathbf{y}} = P^\top \mathbf{c}'$ and likelihood covariance $\hat{\Sigma}_{\hat{\mathbf{y}}} = P^\top C' P$ to project from the old feature space into the new feature space by passing back through data space. The choice that minimizes reconstruction error is the pseudo-inverse $P = (K'_{\mathbf{v'u'}}K'_{\mathbf{u'v'}})^{-1}K'_{\mathbf{v'u'}}$, but requires storage of the full dataset, $(X'_{\text{batch}}, \mathbf{y}')$. Instead we take $P = K'^{-1}_{\mathbf{u'u'}}$, resulting in the following modifications to Eq. (4):

$$\mathbf{c} = K_{\mathbf{uv}}\Sigma_{\mathbf{y}}^{-1}\mathbf{y} + K_{\mathbf{uu}'}K'^{-1}_{\mathbf{u'u'}}\mathbf{c}', \tag{6}$$

$$C = K_{\mathbf{uv}}\Sigma_{\mathbf{y}}^{-1}K_{\mathbf{vu}} + K_{\mathbf{uu}'}(K'^{-1}_{\mathbf{u'u'}}C'K'^{-1}_{\mathbf{u'u'}})K_{\mathbf{uu}'}, \tag{7}$$

Note that $K'^{-1}_{\mathbf{u'u'}}\mathbf{c}' = \Sigma_{\mathbf{y'}}^{-1}\mathbf{y}'$ and $K'^{-1}_{\mathbf{u'u'}}C'K'^{-1}_{\mathbf{u'u'}} = \Sigma_{\mathbf{y'}}^{-1}$ in the special case where $X'_{\text{batch}} = Z'$. We also want to emphasize that although we have only considered two batches of data for the sake of clarity, the approach applies to any number of incoming batches.

**The pseudo-data view:** The above update is equivalent to having an SGPR model with a Gaussian likelihood with covariance $\Sigma = \texttt{blkdiag}(\Sigma_{\hat{\mathbf{y}}}, \Sigma_{\mathbf{y}})$ on the data $\{\texttt{cat}(Z', X_{\text{batch}}), \texttt{cat}(\hat{\mathbf{y}}, \mathbf{y})\}$, where

$$\hat{\mathbf{y}} = K'_{\mathbf{u'u'}}C'^{-1}\mathbf{c}', \quad \Sigma_{\hat{\mathbf{y}}}^{-1} = K'^{-1}_{\mathbf{u'u'}}C'K'^{-1}_{\mathbf{u'u'}}. \tag{8}$$

This interpretation is reminiscent of prior online variational approaches of Csató and Opper [16] and Opper [56]. That is, in the context of conditioning a SVGP, we can assume that we began with data $\{Z', \hat{\mathbf{y}}\}$ and are now observing the new data $\{X_{\text{batch}}, \mathbf{y}\}$. See Appendix C.2 for a more details.

**Extending to SVGPs:** SGPR computes $\mathbf{m_u}$ and $S_{\mathbf{u}}$ as a function of $c$ and $C$ in Eq. (5). However the equations can be reversed to solve for $c$ and $C$ given $\mathbf{m_u}$ and $S_{\mathbf{u}}$, allowing us to condition *any* variational sparse GP into an SGPR model, without touching any previous observations due to the

---

[4] For full generality, we allow the hyper-parameters to vary; however, in our BO experiments, we only consider varying the inducing points as that's all we need to update when computing acquisition functions.

[5] Cached computations that depend on $(X'_{\text{batch}}, \mathbf{y}')$ are highlighted in blue.

conditional independence assumptions of variationally sparse GPs.[6] Note that if the variational parameters are not at the optimal solution when the variational distribution is projected back to the pseudo-data, the projection will be to the targets and likelihood *for which the current variational parameters would be optimal*, which may not correspond well to the data that originally created the model. This potential pitfall is mitigated if the variational parameters are well optimized and is offset by the practical advantages of SVGPs.

**Connection to O-SGPR [9]:**    Formally, the updates described in Eqs. (6) and (7) are equivalent to the O-SGPR approach of Bui et al. [9], as we show in Appendix C.2. The original derivation of O-SGPR is very technical, and does not highlight the similarities between the batch and online SGPR variants. Both the projection and pseudo-data views we have just described provide a much more intuitive way to reason about the behavior of O-SGPR models. Our formulation also eliminates a matrix subtraction operation, which is beneficial for numerical stability.

## 3.2  Inducing Point Selection

Here, we describe inducing point selection during the conditioning procedure to enable better variance reduction on new inputs. While heuristics including re-sampling [9] and data sufficient statistics [35] have been proposed, they either require the number of inducing points to grow or gradually forget old observations. We show in Appendix C.4 that relying exclusively on gradient-based optimization of inducing locations works very poorly in the online setting.

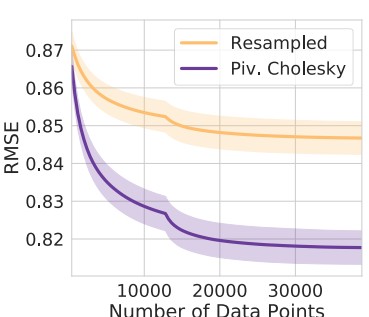

To update the inducing point locations during conditioning, we extend Burt et al. [11]'s batch inducing point initialization approach to heteroskedastic Gaussian likelihoods. They consider theoretical bounds on the marginal likelihood, finding that for homoscedastic Gaussian likelihoods a good strategy is to minimize the trace of the error of a rank-$p$ Nyström approximation, e.g. $\varepsilon = tr\left(\Sigma_{\mathbf{y}}^{-1/2}(K_{\mathbf{vv}} - K_{\mathbf{vu}}K_{\mathbf{uu}}^{-1}K_{\mathbf{uv}})\Sigma_{\mathbf{y}}^{-1/2}\right) = \sigma^{-1/2}tr(K_{\mathbf{vv}} - Q_{\mathbf{vv}})$ for $Q_{\mathbf{vv}} = K_{\mathbf{vu}}K_{\mathbf{uu}}^{-1}K_{\mathbf{uv}}$. They follow a classical approach of Fine and Scheinberg [23] by a greedy minimiziation strategy: choosing as inducing points the pivots of a rank $p$ pivoted Cholesky factorization of $K_{\mathbf{vv}}$.

Figure 2:  Incremental learning RMSE on the UCI *protein* dataset. Pivoted cholesky initialization outperforms resampling.

We denote the function values over the batch+pseudo dataset as $\hat{\mathbf{v}} = [f(\mathbf{x}_1), \ldots, f(\mathbf{x}_b), f(\mathbf{z}_1'), \ldots, f(\mathbf{z}_p')]^\top$. In our case the covariance of the pseudo-likelihood is no longer homoscedastic, so the slack term becomes $\varepsilon = tr(\Sigma^{-1/2}(K_{\hat{\mathbf{v}}\hat{\mathbf{v}}} - Q_{\hat{\mathbf{v}}\hat{\mathbf{v}}})\Sigma^{-1/2})$ and hence the pivoted Cholesky decomposition is instead performed over $\Sigma^{-1/2}K_{\hat{\mathbf{v}}\hat{\mathbf{v}}}\Sigma^{-1/2}$ to select the top $p$ pivots of the $p + n_{\text{new}}$ matrix. When compared to re-sampling the inducing points [9], pivoted cholesky updates perform significantly better, as shown in Figure 2 on the UCI protein dataset [19]. Experimental details are given in Appendix D.2.

**Application to Bayesian Optimization:**    In the context of BO, we condition on *hypothetical* observations, and the conditioned surrogates are discarded after each acquisition function evaluation. Since the SVGP will not be conditioned on more than a few batches of observations, we can sidestep the issue of updating inducing locations entirely by instead conditioning into an *exact GP* trained the combined pseudo-data through the pseudo-likelihood. That is, we model the data as $(\mathbf{y}, \hat{\mathbf{y}}) \sim \mathcal{N}(f, \Sigma)$ (Gaussian with block-diagonal covariance) assuming $f \sim \mathcal{GP}(\mu_\theta, k_\theta(\cdot, \cdot))$. We reach the same model by choosing $\hat{X}$ as the inducing points in our conditioned SGPR (Section 3.1). For small $n_t$, an exact GP is not much slower, taking only $(n_t + p)^3$ computations instead of $p^3$ computations, further reduced by using low rank updates.

## 3.3  Local Laplace Approximations for Non-Gaussian Observations

Thus far, we have solely considered Gaussian observations. The introduction of a non-Gaussian likelihood presents a new challenge, since it implies that the current observation batch and the

---

[6]Alternatively, we could construct SVGPs via direct optimization of $c$ and $C$.

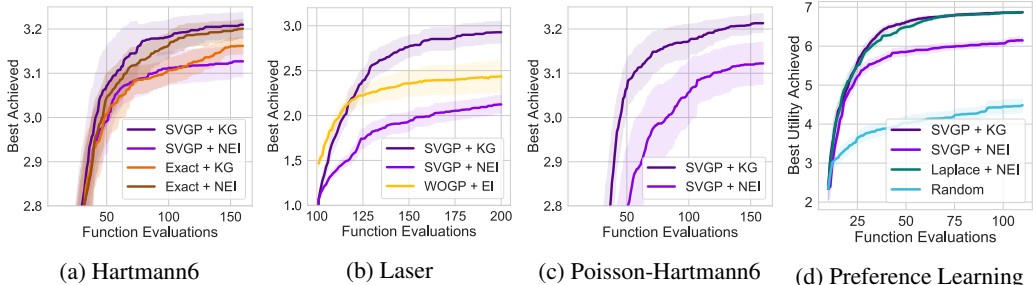

(a) Hartmann6      (b) Laser      (c) Poisson-Hartmann6      (d) Preference Learning

Figure 3: **(a)** Hartmann6 test problem with one constraint. Here, SVGPs with noisy expected improvement (qNEI) and KG match the performance of exact GPs with qNEI and qKG. **(b)** Free electron laser problem from McIntire et al. [51]; SVGPs with knowledge gradient outperforming weighted sparse GPs. **(c)** Constrained Hartmann6 test problem with count responses (Poisson likelihood). Only SVGPs can be used here, and qKG outperforms qNEI. **(d)** Preference learning; SVGPs with qKG are similar to Laplace approximations with NEI, and outperform qNEI with SVGPs.

pseudo-data are no longer jointly Gaussian. To adapt the conditioning procedure to the non-Gaussian setting, we can simply perform a Laplace approximation of the likelihood at the new points [65, Ch. 3]. Specifically, this gives us an approximate likelihood, $\hat{p}(y|f) = \mathcal{N}(\tilde{y}; f, \mathcal{H}_*^{-1})$, where $\mathcal{H}_* = \nabla_f^2 \log p(y|f)|_{f^*(y)}$ and $f^*(y)$ is the maximizer of $\log p(y|f) + f^\top K^{-1} f$, computed via Newton iteration.[7] When conditioning on new observations $\mathbf{y}$, we substitute $f^*(\mathbf{y})$ instead. That is, our new model has responses $(f^*(\mathbf{y}), \hat{\mathbf{y}})$ instead of $(\mathbf{y}, \hat{\mathbf{y}})$, and the pseudo-likelihood remains Gaussian with covariance $\Sigma = \text{blkdiag}(\mathcal{H}_*^{-1}, \Sigma_{\hat{\mathbf{y}}})$. We primarily consider natural parameterizations of one-dimensional exponential families, so that $\mathcal{H}_*$ is positive, diagonal and depends solely on $f$. Computing $\nabla \log p(y|f)$ and $\mathcal{H}_*$ is possible by hand but one can also use automatic differentiation [AD, 63].[8] In Appendix D, Figure 8 we show the effect of repeated Laplace approximations across several batches for online classification.

## 4   Experiments

Our experimental evaluation demonstrates that SVGPs using OVC can be successfully used as surrogate models with advanced acquisition functions in Bayesian optimization, even in the large batch and non-Gaussian settings. In keeping with the BO literature, we will refer to the query batch size as $q$ (not to be confused with the variational posterior $q(f)$ in previous sections). *All SVGP models that use conditioning (or fantasization) require OVC to even be practical to implement.*

**Using OVC as a Building Block inside of BO**

In all of our experiments, we use OVC as a *building block* to enable fantasization (Algorithm 1) within a standard BO acquisition function that requires fantasiziation. These acquisitions are generally "look-ahead" as a result; specifically, qKG [2, 39], LTSs, our version of qGIBBON which uses a fantasy batch [54], and qMultiStepLookAhead [39] all use the fantasization model. After adding in OVC as the `condition_on_observations` function within a BoTorch model class [2], we can simply optimize qKG or qGIBBON with an SVGP exactly as an exact GP surrogate, by using gradient based optimizers such as L-BFGS-B. In general, we need to differentiate through the fantasy model with respect to the inputs and then use gradient based methods to find the optimum. Please see Balandat et al. [2] and Frazier [24] for description of how a BO loop is constructed and how acquisition functions are optimized.

**Experimental Setup**    In general, a Bayesian optimization loop consists of the steps of training the model and then using the trained model to optimize an acquisition function to acquire new data points,

---

[7]One could consider using the posterior covariance instead of $K$. Our experiments with the posterior covariance produced more extreme values of $f$ and thus less regularization.

[8]Specifically, we use PyTorch's functional API, `https://pytorch.org/docs/stable/autograd.html#functional-higher-level-api`

which are then added into the training data for the next model. All experiments use PyTorch [62], GPyTorch [25], and BoTorch [2]. Unless otherwise specified, we run each experiment 50 times and report the mean and two standard deviations of the mean.

In the first step, we first train the inducing points, variational distribution, and kernel hyper-parameters using the evidence lower bound given in Eq. A.3. As all components are differentiable, we use the Adam optimizer with a learning rate of 0.1 and optimize for 1000 steps or until the loss converges, whichever is shorter. To initialize the inducing points, we compute a pivoted cholesky factorization on the initial kernel on the training data (described in Section 3.2 following Burt et al. [11]). The kernel hyper-parameters are initialized to GPyTorch defaults (which sets all lengthscales to one), while the variational distribution is initialized to $\mathbf{m_u} = 0$, $S_{\mathbf{u}} = I$ (again, GPyTorch defaults). Further experimental details and dataset descriptions are in the Appendix.

## 4.1 Knowledge Gradient with SVGPs

These experiments use the one-shot formulation of the batch knowledge gradient (qKG) (Eq. 3) from Balandat et al. [2], who demonstrated that qKG outperforms other acquisitions due to being able to plan two steps into the future. *Using and optimizing qKG has only been available for exact GPs previously.* By using OVC, we have enabled SVGPs to also efficiently and tractably optimize qKG, even for non-Gaussian observations. We compare to batch noisy expected improvement [qNEI, 45] which is myopic and does not use fantasization (e.g. conditioning). Here, for the SVGPs we used $\min(N, 25)$ inducing points.

**Gaussian observations:** We use the **Hartmann6** test function, with one black box constraint, maximizing $f(x) = -\sum_{i=1}^{4} \alpha_i \exp\{-\sum_{j=1}^{6} A_{ij}(x_j - P_{ij})^2\}$ subject to the constraint that $c(x) = \|x\|_1 \leq 3$ for fixed $A, P, \alpha$. We use 10 initial points and a batch size of 3 optimizing for 50 iterations, comparing to SVGPs and exact GPs using qNEI. We show the results in Figure 3a where SVGPs with qKG match exact GPs with both qNEI and qKG, and outperform SVGPs using qNEI.

Second, we mimic the **laser tuning** experiment of [51, 20], demonstrating that SVGPs outperform even weighted online GPs (WOGP), which were designed for this task. Here, we use 100 initial points, with $d = 14$, and and wish to tune a laser's output energy as a function of the magnet settings that produce the beam. Like McIntire et al. [51] we treat a pretrained GP fit on experimental data as a simulator. We use a batch size of 1, finding that SVGPs + KG outperform WOGP (Figure 3b). However, exact GPs outperform the variational approaches (Appendix Fig. 9b).

**Non-Gaussian likelihoods:** Next, we extend the knowledge gradient to problems with non-Gaussian likelihoods. First, we take the constrained Hartmann6 test function from the previous section, and use **Poisson** responses, $y \sim \text{Poisson}(\exp\{f(x)\})$, repeating the same settings as for the Gaussian case. Now, the data is non-Gaussian and cannot be well-modelled by a Gaussian likelihood, so we compare to only SVGPs with qNEI. qNEI is outperformed by qKG, as shown in Figure 3c.

Second, in Figure 3d, we consider a **preference learning** problem inspired by Lin et al. [46]. Here, the latent data is described by $f(x) = -10^{-1/2} \sum_{i=1}^{10} \sqrt{i} \cos(2\pi x_i)$ for $x \in [0, 1]^{10}$, comparing to Laplace approximations [15]. Again, we see that SVGPs with qKG outperform qNEI with both SVGPs and Laplace approximations.

## 4.2 Active Learning of Disease Incidence

We next present results for two active learning tasks governing the collection of disease incidence data. In both tasks the acquisition functions again require efficient conditioning on hypothetical data, and the second task has Binomial responses, so exact GPs cannot be applied. In both settings, applying OVC to SVGPs gives strong results competitive with either exact GPs or random forests.

**Modelling of Malaria Incidence:** We consider data from the Malaria Global Atlas [81] describing the infection rate of a parasite known to cause malaria in 2017. We wish to choose spatial locations to query malaria incidence in order to make the best possible predictions on a withheld test set, the entire country of Nigeria. Following Stanton et al. [70], we minimize the negative integrated posterior variance [NIPV, 68], defined as $a(\mathbf{x}; \mathcal{D}) := -\int_{\mathcal{X}} \mathbb{E}(\mathbb{V}(f(\mathbf{x})|\mathcal{D}_{+\mathbf{x}})|\mathcal{D})d\mathbf{x}$, again with $\mathcal{D}_{+\mathbf{x}} = \mathcal{D} \cup \{(\mathbf{x}, y)\}$. Intuitively, the minimizer of this acquisition will be the batch of data points that when added into the model will most reduce the total posterior uncertainty across the domain,

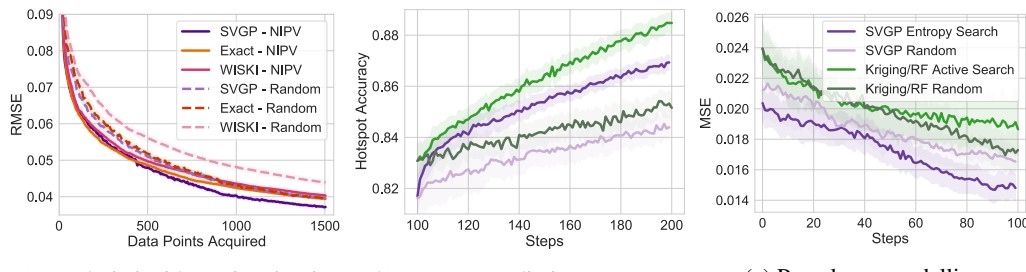

(a) Malaria incidence in Nigeria      (b) Hotspot prediction accuracy      (c) Prevalence modelling

Figure 4: **(a)** Active learning of malaria incidence from satellite data. Using qNIPV outperforms randomly selecting points, while the SVGPs slightly outperform both exact GPs and WISKI. **(b,c)** Active learning of schistomiasis incidence in Cote d'Ivoire from Andrade-Pacheco et al. [1]. Comparison is to the random forest based approach using active sampling. While the GP based models are somewhat less accurate at predicting hotspots **(b)**, they are better as a global model of prevalence **(c)**.

requiring efficient conditioning to do so in a tractable manner. The results are shown for a batch size of $q = 6$ across 15 trials in Figure 4a where we see that each method outperforms random baselines. Perhaps due to the optimization freedom, the SVGP outperforms both the exact GP and WISKI [70].

**Hotspot Modelling:** We follow Andrade-Pacheco et al. [1] and model the prevalence of schistomiatosis in Côte d'Ivoire using simulated responses from 1500 villages in that country, and taking into account six other demographic variables. We model the responses $y$ (incidence) at locations $\mathbf{x}$ with population $n(\mathbf{x})$ with a Binomial likelihood $p(y|f, \mathbf{x}) \sim \text{Binomial}(n(\mathbf{x}), r(f))$, where $r(f) = (1 + \exp\{-f\})^{-1}$. Letting $\tau \in (0, 1)$ be a threshold on the prevalence for a location to be considered a "hotspot" [1], we compute the entropy:

$$h_\tau(\mathbf{x}, \mathcal{D}) := \mathbb{E}_{p(f|\mathcal{D})}(\mathbb{H}(\text{Bernoulli}(f > \text{logit}(\tau)))) \approx \frac{1}{K} \sum_{i=1}^{K} 1_{f > \text{logit}(\tau)} \mathbb{H}(\text{Bernoulli}(f)),$$

taking the acquisition value to be the reduction in the entropy of the posterior predictive distribution over the incidence under the hotspot-focused likelihood.

$$a_\tau(\mathbf{x}, \mathcal{D}) := \int_{\mathbf{x}' \in \mathcal{X}} \left( h_\tau(\mathbf{x}', \mathcal{D}_{+\mathbf{x}}) - h_\tau(\mathbf{x}', \mathcal{D}) \right) d\mathbf{x}'. \tag{9}$$

A location is given a high acquisition value if observing the incidence at that location reduces the uncertainty of the model on the predicted set of hotspots. In Figure 4b, we compare to Andrade-Pacheco et al. [1] who use spatial kriging on the residuals of a random forest model. Both their random baseline and their exploration based procedure (a variant of UCB) start off with higher prediction accuracy; however our SVGP models ultimately outperform the kriging approach with random selection. The SVGP is a better predictor of true prevalence, as shown in Figure 4c. In both cases, our acquisition function significantly outperforms random selection with a SVGP surrogate.

### 4.3 Rollouts within Thompson Sampling for High Dimensional BO

For our final set of experiments, we solve control problems using trust region Bayesian optimization [TurBO, 22]. Inspired by multi-step look-aheads [39, 4], we propose $h$-step look-ahead Thompson sampling (LTS-$h$). In BO, Thompson sampling (TS) is often implemented by drawing samples from the posterior at points all over the domain, then selecting the $q$ best to form a query batch [71, 22]. LTS-$h$ extends the idea by conditioning the surrogate independently on each posterior sample in the original TS query batch, Thompson sampling again from the updated posterior with a new set of points, and appending the best sample to its predecessor to form a *path*. The process is repeated $h$ times. Finally, we condition the original surrogate jointly on each path, then perform TS again to choose the new query batch. Informally, each path corresponds to a distinct, coherent draw from $p(f|\mathcal{D})$, allowing the inner-loop to refine its guess of the global optimum for different $f$, and the final round of TS chooses the query batch based on those guesses. See Appendix C.5 for a formal description. Like other look-ahead acquisitions, LTS-$h$ is only practical if posterior conditioning and samples are very efficient and numerically stable. LTS-$h$ is conceptually similar to path sampling for look-ahead in Jiang et al. [39] and kriging believer [26].

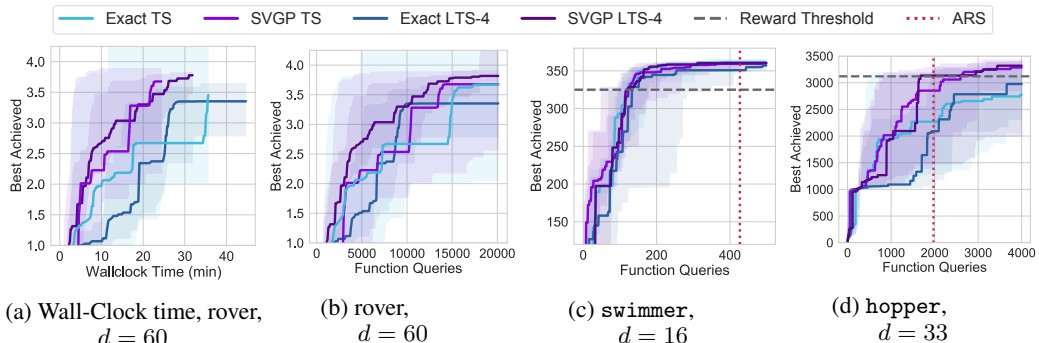

Figure 5: Multi-step rollouts with OVC and SVGPs provides sample-complexity and wall-clock time improvements on high dimensional BO problems when using TurBO and LAMCTS [22, 78]. SVGP rollouts are as time efficient (to 150 iterations) as standard TS **(a)** on lunar rover, $d = 60$. **(c-d)** MuJoCo environments using LAMCTS + TurBO. Also shown is the reward threshold (dashed grey lines) and augmented random search (ARS)'s performance (dotted red lines) [49]. The median and its $95\%$ confidence interval are shown over 24 trials for rover and 10 trials for swimmer and hopper.

For validation, we consider tuning the 60 dimensional path that a **lunar rover** takes across a field stacked with obstacles [79, 22]. We use batches of $q = 100$ with 200 initial points, and use trust region Bayesian optimization (TurBO) to split up the space effectively [22], comparing to TurBO with Thompson sampling (TS) as the acquisition [71]. We show the wall clock times per iteration in Figure 5a, where we see that the TurBO + LTS approaches compare well in wallclock time to TurBO+TS, while being more function efficient (Figure 5b).

Finally, we consider **MuJoCo** problems using the OpenAI gym [75, 7] with LTSs inside of TurBO with trust regions generated by Monte Carlo Tree Search following the procedure of Wang et al. [78]. Following Wang et al. [78], we learn a linear policy and consider the `swimmer-v2, hopper-v2` environments over 10 trials displaying the median and its $95\%$ confidence band due to high variance. On both problems, SVGP with LTSs tend to be the most sample efficient, with SVGP + TS performing at least as well on `swimmer-v2`. We also show the reward threshold and the performance achieved by augmented random search (ARS), which is a strong baseline reinforcement learning method that uses random search to tune linear controllers [49]. Our results suggest that LTSs are promising overall; however, more work needs to be done for high dimensional kernels on these problems.

## 5 Discussion

In conclusion, we have demonstrated how to efficiently condition on new data points with stochastic variational Gaussian processes via closed form updates to the variational distribution. Our conditioning approach generalizes exact GP conditioning via Laplace approximations for non-Gaussian likelihoods. As a result, we have decoupled look-ahead BO acquisition functions from their dependence on exact GP inference through a Gaussian likelihood, increasing the range, scale, and efficiency of BO applications. In the future, we hope to extend OVC to multi-task and deep Gaussian processes for use in Bayesian optimization [31, 17].

## Acknowledgements

WJM, SS, AGW are supported by an Amazon Research Award, NSF I-DISRE 193471, NIH R01 DA048764-01A1, NSF IIS-1910266, and NSF 1922658 NRT-HDR: FUTURE Foundations, Translation, and Responsibility for Data Science. WJM was additionally supported by an NSF Graduate Research Fellowship under Grant No. DGE-1839302. SS is additionally supported by the United States Department of Defense through the National Defense Science & Engineering Graduate (ND-SEG) Fellowship Program. We'd like to thank Greg Benton for setting up the volatility experiment and for helpful discussions and Nate Gruver and Eytan Bakshy for helpful comments.

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
