# Supplementary Materials for Conditioning Sparse Variational Gaussian Processes for Online Decision-making

## Table of Contents

The Appendix is structured as follows:

- Appendix C discusses Newton's method and training objectives for the variational models, before giving more detail on the methodological work, including implementation details.

## A  Limitations and Societal Impacts

### A.1  Limitations

From a practical perspective, we see several interrelated limitations:

- In our software implementation, we currently support single-output functions. This is partly because GPyTorch currently has limited support for fantasization of multi-task Gaussian processes; see `https://github.com/cornellius-gp/gpytorch/pull/805`.

- For non-Gaussian likelihoods, the approximation may break down for long rollout time steps due to accumulating error from the Laplace approximation.

- Incorrect modelling and thus incorrect rollouts will tend to be even more influential in high dimensional settings, as the kernels we use do not tend to work particularly well in high dimensions [22].

- If the underlying data is non-stationary, then long range predictions may suffer. For example, local models are superior on the rover problem in Figure 5, even compared to global models with advanced acquisition functions (e.g. Figure 12c). This limitation is remedied somewhat by the local modelling approaches we use [22, 78].

- Using many inducing points can worsen numerical conditioning, leading to less stable multi-step fantasization, despite the theoretical advantages of more inducing points [e.g., 3].

### A.2  Societal Impacts

We do not anticipate that our work will have negative societal impacts. To the contrary, as we demonstrate in this paper, Bayesian optimization is naturally suited to applications in the public interest, such as public health surveillance [1]. These types of applications should help benefit global populations by allowing more targeted interventions in the public health setting. However, reliance solely on machine learning models, for example, in quantitative finance settings (as our volatility model in Figure 1c is designed for), could potentially lead to over-confidence and financial shocks as has previously been the case [48].

# B  Further Background

In this section, we describe further related work on both variational inference in the streaming setting as well as the use of sparse GPs in both BO and control before describing Newton's iteration for Laplace approximations and training methods for exact and variational GPs.

## B.1  Extended Related Work

Because our work explores three distinct applications, namely black-box optimization, active learning, and control, there was more noteworthy related work than the space constraints of the main text would permit. Here we present a more complete literature review.

We do not focus on other approaches for streaming variational GP inference ranging from decoupled inducing points [13, 40, 69] to alternative variational constructions [52, 53] to Kalman filtering based approaches [37, 36] to deep linear model approaches [59, 74]. These approaches could potentially be folded into OVC; however, we leave exploration of these for related work.

We also focus solely on the variationally sparse GPs introduced by Titsias [73] and Hensman et al. [33], rather than other approaches, which are potentially amenable to being used within OVC. Of particular note, the classical variational approximations for non-Gaussian likelihoods such as the approaches proposed originally by Csató and Opper [16] and later Opper and Archambeau [57] seem particularly promising, as is the development of kernel methods within the exponential family more generally [12].

From a software perspective, both GPFlowOpt [42] and its successor Trieste (`https://github.com/secondmind-labs/trieste`) seem to support variational GPs for BO but we do not know of a comprehensive benchmarking of their implementation. Implementing variational GPs for most acquisitions in BoTorch [2] is entirely possible using GPyTorch, as we did, although it requires some engineering work and is not natively supported at this time.

Recent work on molecular design has combined variational auto-encoders (VAEs) trained to map high dimensional structured molecular representations to low-dimensional latent representations with SVGPs trained to predict a molecular property of interest from the latent encoding [28, 76], but they primarily use simple acquisition functions.

In the control literature, sparse GPs are more popular, stemming from the seminal works of Csató and Opper [16] and Girard et al. [27]. Chowdhary et al. [14] directly apply the approach of Csató and Opper [16] to optimal control problems, while Ling et al. [47] sparsify GPs for active learning and planning. Groot et al. [29], Boedecker et al. [6], and Bijl et al. [5] perform multiple time step look ahead via moment matching using sparse GPs, while Pan et al. [60] use a similar approach with random fourier features. Deisenroth and Rasmussen [18] use sparse GPs to speed up dynamics models for robotics, while Sæmundsson et al. [66] use sparse GPs for meta reinforcement learning. Finally, Xu et al. [85] use sparse GPs for robot localization tasks in control.

## B.2  Laplace Approximations and Newton's Iteration

Newton iteration iterates

$$f_{t+1} = (K^{-1} + W)^{-1}(Wf_t + \nabla \log p(y|f_t))$$
$$= (K - KW^{-1/2}B^{-1}W^{-1/2}K)(Wf_t + \nabla \log p(y|f_t)), \qquad B := (I + W^{1/2}KW^{1/2}),$$

with $W = -\nabla_f^2 \log p(y|f)$, e.g. the negative Hessian of the log likelihood, until $f_{t+1} - f_t < \epsilon$ (e.g. until a stationary point is reached). We implemented a batch version of the convergence rule, stopping when the total difference is under a threshold for all batches of $f$. The expensive computational cost is that of the solve of $B$, as $W$ is diagonal. In our case, we only consider Newton iteration on test points, so that the time complexity is just $\mathcal{O}(n_{\text{test}}^3)$. Please see Rasmussen and Williams [Chapter 3 of 65] for further detail.

Implementation wise, for all but the GPCV experiment in Figure 1c, we manually implemented the gradient and Hessian terms as they are well known due to being natural parameterizations of exponential families. For the preference learning experiment, we followed the gradient derivation in Chu and Ghahramani [15].

## B.3 Training Mechanisms for Exact Gaussian Processes and Variational Gaussian Processes

Please see the more detailed summaries of Rasmussen and Williams [65] for training methods of exact Gaussian processes as well as the theses of Van der Wilk [77] and Matthews [50] for training methods of sparse Gaussian processes. In what follows, we assume solely a zero mean function and suppress dependence on $\theta$ for the kernel matrices.

**Log Marginal Likelihood for Exact GPs** Training the GP's hyper-parameters, $\theta$, proceeds via maximizing the marginal log-likelihood (MLL). The MLL is given by

$$\log p(\mathbf{y}|X, \theta) = -\frac{1}{2}\mathbf{y}^\top (K_{\mathbf{vv}} + \sigma^2 I)^{-1}\mathbf{y} - \frac{1}{2}\log|K_{\mathbf{vv}} + \sigma^2 I| - \frac{n}{2}\log 2\pi,$$

over the training set, $\mathcal{D} = (X, \mathbf{y})$.

**Collapsed Evidence Lower Bound for SGPR** Sparse Gaussian process regression (SGPR) begins by approximating the kernel with a lower rank version. The training data covariance, $K_{\mathbf{vv}}$, is replaced by the Nyström approximation $K_{\mathbf{vv}} \approx Q_{\mathbf{vv}} := K_{\mathbf{vu}}K_{\mathbf{uu}}^{-1}K_{\mathbf{uv}}$, where $K_{\mathbf{uu}}$ is the kernel evaluated on the inducing point locations, $Z$. SGPR learns the locations of the inducing points and the kernel hyperparmeters through a 'collapsed' form of the evidence lower bound (ELBO), yielding a variational adaptation of older Nyström or projected process approximations [65, Chapter 7].

The ELBO is called "collapsed" because the Gaussian likelihood allows the parameters of the variational distribution to be analytically optimized and integrated out (collapsed), yielding a bound that depends only on the inducing point locations and the kernel hyperparameters [73]. The SGPR bound is as follows (see Titsias [72] for the full derivation):

$$\mathcal{F}(\theta, Z) := \log p(y|0, \sigma^2 I + Q_{\mathbf{vv}}) - \frac{1}{2\sigma^2}\text{trace}(K_{\mathbf{vv}} - Q_{\mathbf{vv}}). \tag{A.1}$$

We can apply standard gradient based training to both the kernel hypers $\theta$ as well as the inducing locations $Z$. Jankowiak et al. [38] derive a variational version of this bound which enables subsampling across data points; we leave training with that bound for future work.

**Evidence Lower Bound for SVGP** The advance of Hensman et al. [33] is that they do not compute the optimal variational parameters at each time step proposing the sparse variational GP (SVGP). Their derivation, see Hensman et al. [33, 34] for a full derivation, yields an 'uncollapsed' ELBO (so named due to its explicit dependence on the variational parameters),

$$\mathcal{F}(\theta, Z, \mathbf{m}_\mathbf{u}, S_\mathbf{u}) := \mathbb{E}_{q(f)}\log p(y|f) - \text{KL}(\phi(\mathbf{u})||p(\mathbf{u})), \tag{A.2}$$

where $q(f) = \int p(f|\mathbf{u})\phi(\mathbf{u})d\mathbf{u}$. The latent GP $f$ is replaced with the variational distribution, $\phi(\mathbf{u}) = \mathcal{N}(\mathbf{m}_\mathbf{u}, S_\mathbf{u})$. $q(f)$ can be determined via Gaussian marginalization from $\phi(\mathbf{u})$. and is $q(f) = \mathcal{N}(K_{\mathbf{vu}}K_{\mathbf{uu}}^{-1}\mathbf{m}_\mathbf{u}, K_{\mathbf{vv}} - K_{\mathbf{vu}}K_{\mathbf{uu}}^{-1}(K_{\mathbf{uu}} - S_\mathbf{u})K_{\mathbf{uu}}^{-1}K_{\mathbf{uv}})$. Mini-batching is possible because the first term in Eq. A.2 splits over each of the $n$ data points as each $y_i$ is conditionally independent given $f$, allowing sub-sampling. Sub-sampling and mini-batching enable the use of stochastic optimization techniques, reducing the per iteration cost to be constant in $n$, the number of data points.

Bui et al. [9] introduce a streaming version of the ELBO that involves two further terms; the method is called O-SVGP. Their primary model, O-SGPR (called the "collapsed bound" of a streaming sparse GP in that paper) is trained through a variant of their ELBO bound that integrates out the variational parameters. We describe this collapsed bound in more detail in Appendix C.4.

**Predictive Log Likelihood for SVGP** We close this section by quickly describing the predictive log likelihood (PLL) method of Jankowiak et al. [38], which is motivated by attempting to improve the calibration of SVGP models trained via the ELBO. The key advance in Jankowiak et al. [38] is that they consider the expectation over both the data and the response, rather than simply the expectation over the response in the variational distribution, producing an objective that becomes:

$$\mathcal{F}(\theta, x_m, \mathbf{m}_\mathbf{u}, S_\mathbf{u}) := \mathbb{E}_{p(y,x)}\log p(y|f, x) - \text{KL}(\phi(\mathbf{u})||p(\mathbf{u}))$$

$$\approx \sum_{i=1}^{N}\log\left(\mathbb{E}_{\phi(\mathbf{u})p(f_i|\mathbf{u},x_i)}p(y_i|f_i)\right) - \text{KL}(\phi(\mathbf{u})||p(\mathbf{u})).$$

We consider this optimization objective for several of the larger-scale problems here.

## C  Further Methodological Details

In this Appendix, we begin by presenting our approach, OVC, as a generalization of exact Gaussian conditioning for SGPR (C.1) before describing an alternative interpretation of Bui et al. [9] that is equivalent to our approach in C.2. Then, in Appendix C.3, we describe the practical implementation of OVC. In C.4, we describe a flaw of the O-SGPR bound for small batch sizes. Finally, in C.5, we give an extended description of look-ahead Thompson sampling (LTS).

### C.1  OVC Generalizes Efficient Batch SGPR Conditioning

In this section, we show that OVC can also be viewed as a generalization of Gaussian conditioning for SGPR. Gaussian conditioning for SGPR is recovered as a special case of OVC when $\theta' = \theta$ and $Z' = Z$. Under this assumption, Eqns. (6) and (7) simplify as follows:

$$\mathbf{c} = K_{\mathbf{uv}}\Sigma_{\mathbf{y}}^{-1}\mathbf{y} + \mathbf{c}', \quad C = K_{\mathbf{uv}}\Sigma_{\mathbf{y}}^{-1}K_{\mathbf{vu}} + C'. \tag{A.3}$$

Considering the predictive distribution given by Eq. 2 in the main text, as we add new data points, $(X_{\text{batch}}, \mathbf{y})$, into the model, we need to update $A = (K_{\mathbf{uu}} + C)^{-1}$, $\mathbf{a} = A\mathbf{c}$, and $R = (K_{\mathbf{uu}}^{-1} - A)^{1/2}$. For homoscedastic likelihoods the $A$ update is a fast low-rank update via the Woodbury matrix identity,

$$A = A' - A'K_{\mathbf{uv}}(\sigma^2 I + K_{\mathbf{vu}}A'K_{\mathbf{uv}})^{-1}K_{\mathbf{vu}}A'. \tag{A.4}$$

This produces efficient Sherman-Morrison updates to generate $\mathbf{a}$ (via addition and matrix vector multiplication) and Woodbury based updates to update $R$, the predictive covariance cache via low-rank updates (e.g. Proposition 2 of Jiang et al. [39]).

Furthermore, these low rank updates can be used to produce updates to the exact caches. These updates are simply exact Gaussian conditioning with an approximate kernel. That is, the SGPR caches are merely transformed exact caches for any given set of data points. To demonstrate, we use a Nyström approximation for the kernel throughout, e.g. $K_{\mathbf{vw}} \approx K_{\mathbf{vu}}K_{\mathbf{uu}}^{-1}K_{\mathbf{uw}}$, then after applying Woodbury we can write the exact caches using $A$:

$$RR^{\top} = (K_{\mathbf{vv}} + \Sigma_{\mathbf{y}})^{-1} = \Sigma_{\mathbf{y}}^{-1} - \Sigma_{\mathbf{y}}^{-1}K_{\mathbf{vu}}(K_{\mathbf{uu}} + K_{\mathbf{uv}}\Sigma_{\mathbf{y}}^{-1}K_{\mathbf{vu}})^{-1}K_{\mathbf{uv}}\Sigma_{\mathbf{y}}^{-1}$$
$$= \Sigma_{\mathbf{y}}^{-1} - \Sigma_{\mathbf{y}}^{-1}K_{\mathbf{vu}}AK_{\mathbf{uv}}\Sigma_{\mathbf{y}}^{-1}$$

with a similar expression for the predictive mean cache as

$$\mathbf{a} = (K_{\mathbf{vv}} + \Sigma_{\mathbf{y}})^{-1}\mathbf{y} = \Sigma_{\mathbf{y}}^{-1}\mathbf{y} - \Sigma_{\mathbf{y}}^{-1}K_{\mathbf{vu}}A\mathbf{c}.$$

Next we take the caches computed on every training point and project them into the space of inducing points by multiplying them by $K_{\mathbf{uu}}^{-1}K_{\mathbf{uv}}$. It takes a bit of algebra, but we can derive updated expressions for $RR^{\top}$ and $\mathbf{a}$ in terms of solely the new covariance matrix, $K_{\mathbf{vu}}$ and the new responses, $\Sigma_{\mathbf{y}}^{-1}\mathbf{y}$. That is, $\mathbf{a}_{\text{SGPR}} = K_{\mathbf{uu}}^{-1}K_{\mathbf{uv}}\mathbf{a}$ and $RR_{\text{SGPR}}^{\top} = K_{\mathbf{uu}}^{-1}K_{\mathbf{uv}}RR^{\top}K_{\mathbf{vu}}K_{\mathbf{uu}}^{-1}$.

$$\mathbf{a}_{\text{SGPR}} = K_{\mathbf{uu}}^{-1}K_{\mathbf{uv}}(\Sigma_{\mathbf{y}}^{-1}\mathbf{y} - \Sigma_{\mathbf{y}}^{-1}K_{\mathbf{vu}}AK_{\mathbf{uv}}\Sigma_{\mathbf{y}}^{-1}\mathbf{y}) = K_{\mathbf{uu}}^{-1}\mathbf{c} - K_{\mathbf{uu}}^{-1}C(K_{\mathbf{uu}} + C)^{-1}\mathbf{c}$$
$$= K_{\mathbf{uu}}^{-1}((K_{\mathbf{uu}} + C)(K_{\mathbf{uu}} + C)^{-1} - C(K_{\mathbf{uu}} + C)^{-1})\mathbf{c} = (K_{\mathbf{uu}} + C)^{-1}\mathbf{c}$$

and similarly

$$RR_{\text{SGPR}}^{\top} = K_{\mathbf{uu}}^{-1}K_{\mathbf{uv}}RR^{\top}K_{\mathbf{vu}}K_{\mathbf{uu}}^{-1} = K_{\mathbf{uu}}^{-1}K_{\mathbf{uv}}(\Sigma_{\mathbf{y}}^{-1} - \Sigma_{\mathbf{y}}^{-1}K_{\mathbf{vu}}AK_{\mathbf{uv}}\Sigma_{\mathbf{y}}^{-1})K_{\mathbf{vu}}K_{\mathbf{uu}}^{-1}$$
$$= K_{\mathbf{uu}}^{-1}K_{\mathbf{uu}} - K_{\mathbf{uu}}^{-1}CACK_{\mathbf{uu}}^{-1} = K_{\mathbf{uu}}^{-1}(K_{\mathbf{uu}}(K_{\mathbf{uu}} + C)^{-1}CK_{\mathbf{uu}}^{-1})$$
$$= (K_{\mathbf{uu}} + C)^{-1}CK_{\mathbf{uu}}^{-1} = (K_{\mathbf{uu}} + K_{\mathbf{uu}}C^{-1}K_{\mathbf{uu}})^{-1}$$
$$= K_{\mathbf{uu}}^{-1} - (K_{\mathbf{uu}} + C)^{-1}.$$

Similarly one could follow this logic in reverse to go from SGPR caching to caching for exact GP inference. We can also use the updates in Eq. A.3 to update the exact GPs caches via first updating the SGPR caches.

To summarize, exact GP regression is just Gaussian conditioning, which can be viewed as a special case of SGPR if one inducing point is placed at every data point. SGPR in turn is again Gaussian conditioning through an approximate kernel on projected features, which can be viewed as a special case of O-SGPR if the inducing points and kernel hyperparameters are held fixed. Finally O-SGPR can be viewed as a special case of O-SVGP if the variational parameters are constrained to be optimal.

## C.2 Interpreting Bui et al. [9] as O-SGPR

The approach outlined in Section 3.1 can be verified to be equivalent to streaming sparse GPs (e.g. the un-collapsed bound of Bui et al. [9]) mechanically by verifying that the expressions for the ELBO are equivalent (up to constants) and that the predictive mean and variance are exactly equivalent. Although verifying the equivalence is simply a matter of manipulating algebraic expressions, we have not yet justified the choice of $\hat{\mathbf{y}}$ and $\Sigma_{\hat{\mathbf{y}}}$. Bui et al. [9] obtained their expressions by means of variational calculus, and arrived at the correct result, but did not provide much in the way of intuition for the nature of the optimal solution.

We now show how the choice of pseudo-targets $\hat{\mathbf{y}}$ and pseudo-likelihood covariance $\Sigma_{\hat{\mathbf{y}}}$ has a clear interpretation that obviates any need to appeal to variational calculus except as a formal guarantee of optimality. Again, suppose we are given a sparse variational GP with inducing points $Z'$, kernel hyperparameters $\theta'$ and a pre-computed optimal variational distribution $\phi(\mathbf{u}') = \mathcal{N}(\mathbf{m}_{\mathbf{u}'}, S_{\mathbf{u}'})$, and then asked to find the likelihood and dataset of size $p$ that produced the model. Although the problem as stated is under-determined, if we choose $X = Z'$ and assume the likelihood is some Gaussian centered at $f$, then we can reverse Eqn. (5) (in the main text) to solve for $\mathbf{y}$ and $\Sigma_{\mathbf{y}}$ as follows:

$$\mathbf{m}_{\mathbf{u}'} = K'_{\mathbf{u}'\mathbf{u}'}(K'_{\mathbf{u}'\mathbf{u}'} + K'_{\mathbf{u}'\mathbf{u}'}\Sigma_{\mathbf{y}}K'_{\mathbf{u}'\mathbf{u}'})^{-1}K'_{\mathbf{u}'\mathbf{u}'}\Sigma_{\mathbf{y}}^{-1}\mathbf{y},$$
$$\Rightarrow \mathbf{y} = (\Sigma_{\mathbf{y}}K'^{-1}_{\mathbf{u}'\mathbf{u}'} + I)\mathbf{m}_{\mathbf{u}'} = \hat{\mathbf{y}} \tag{A.5}$$
$$S_{\mathbf{u}'} = K'_{\mathbf{u}'\mathbf{u}'}(K'_{\mathbf{u}'\mathbf{u}'} + K'_{\mathbf{u}'\mathbf{u}'}\Sigma_{\mathbf{y}}K'_{\mathbf{u}'\mathbf{u}'})^{-1}K'_{\mathbf{u}'\mathbf{u}'},$$
$$\Rightarrow \Sigma_{\mathbf{y}} = (S_{\mathbf{u}'}^{-1} - K'^{-1}_{\mathbf{u}'\mathbf{u}'})^{-1} = \Sigma_{\hat{\mathbf{y}}}. \tag{A.6}$$

As a result, we can now provide new, intuitive interpretations of $\hat{\mathbf{y}}$, $\Sigma_{\hat{\mathbf{y}}}$ and $\phi(\mathbf{u})$. In simple terms, the streaming sparse GP (i.e. O-SGPR) of Bui et al. [9] is equivalent to a sequence of SGPR models, where instead of training on all previously observed data through the original likelihood at each timestep, each model trains *only* on the combination of the current batch of data $(X_{\text{batch}}, \mathbf{y})$, and the pseudo-data $(Z', \hat{\mathbf{y}})$ through a pseudo-likelihood with covariance $\Sigma = \text{blkdiag}(\Sigma_{\hat{\mathbf{y}}}, \Sigma_{\mathbf{y}})$. The pseudo-data and pseudo-likelihood together represent all the past data and models. Furthermore, $(Z', \hat{\mathbf{y}})$ and $\Sigma_{\hat{\mathbf{y}}}$ are the *unique* size-$m$ dataset with $X = Z'$ and $f$-centered Gaussian likelihood that could have produced $\phi(\mathbf{u}')$, given $Z'$ and $\theta'$. In other words we can think of the tuple $(\theta', Z', \hat{\mathbf{y}}, \Sigma_{\hat{\mathbf{y}}})$ as a compressed representation of the sparse GP it defines.

## C.3 Practical Implementation

Implementation wise and to reduce our engineering overhead, we focused on computing $\hat{\mathbf{y}}$ and $\Sigma_{\hat{\mathbf{y}}}$ in a numerically stable manner. We start with the pseudo-covariance term, which can be simplified as

$$\Sigma_{\hat{\mathbf{y}}} = I + S_{\mathbf{u}'}(K'_{\mathbf{u}'\mathbf{u}'} - S_{\mathbf{u}'})^{-1}S_{\mathbf{u}'}.$$

After some more algebra, we can rewrite the pseudo-observations that depend on the inducing points,

$$\hat{\mathbf{y}} = \Sigma_{\hat{\mathbf{y}}}S_{\mathbf{u}'}^{-1}\mathbf{m}_{\mathbf{u}'} = \left(I + S_{\mathbf{u}'}(K'_{\mathbf{u}'\mathbf{u}'} - S_{\mathbf{u}'})^{-1}S_{\mathbf{u}'}\right)S_{\mathbf{u}'}^{-1}\mathbf{m}_{\mathbf{u}'}$$
$$= S_{\mathbf{u}'}^{-1}\mathbf{m}_{\mathbf{u}'} + S_{\mathbf{u}'}(K'_{\mathbf{u}'\mathbf{u}'} - S_{\mathbf{u}'})^{-1}\mathbf{m}_{\mathbf{u}'} \tag{A.7}$$

For numerical stability, we replace inverses of matrix subtractions as $\left((K - S)^{-1}(K - S)^{-\top}\right)(K - S)^{\top}$, dropping subscripts. While there is still a matrix subtraction here, squaring the system improves the numerical stability of the systems, as we are forcing all of the eigenvalues of the matrices that we are solving linear systems with to be non-negative.

In practice however, we use "whitening" of the variational distribution as introduced by [50]. We instead optimize $\bar{\mathbf{m}}_{\mathbf{u}} = K_{\mathbf{u}\mathbf{u}}^{-1/2}\mathbf{m}_{\mathbf{u}}$ and $\bar{S}_{\mathbf{u}} = K_{\mathbf{u}\mathbf{u}}^{-1/2}S_{\mathbf{u}}K_{\mathbf{u}\mathbf{u}}^{-1/2}$. We can rewrite $\Sigma_{\hat{\mathbf{y}}}$ using the whitened variational covariance matrix $\bar{S}_{\mathbf{u}'}$ producing, $\Sigma_{\hat{\mathbf{y}}} = K'^{1/2}_{\mathbf{u}'\mathbf{u}'}(\bar{S}_{\mathbf{u}'} + \bar{S}_{\mathbf{u}'}(I - \bar{S}_{\mathbf{u}'})^{-1}\bar{S}_{\mathbf{u}'})K'^{1/2}_{\mathbf{u}'\mathbf{u}'}$. Again, we square the second term to enhance stability, although it already has a symmetric form; that is, we compute

$$\Sigma_{\hat{\mathbf{y}}} = K'^{-1/2}_{\mathbf{u}'\mathbf{u}'}(\bar{S}_{\mathbf{u}'} + \bar{S}_{\mathbf{u}'}(I - \bar{S}_{\mathbf{u}'})^{-1}(I - \bar{S}_{\mathbf{u}'})^{-\top}(I - \bar{S}_{\mathbf{u}'})^{\top}\bar{S}_{\mathbf{u}'})K'^{-1/2}_{\mathbf{u}'\mathbf{u}'}.$$

Similarly, Eq. A.6 simplifies to become

$$\hat{\mathbf{y}} = K'^{1/2}_{\mathbf{u}'\mathbf{u}'}(I - \tilde{S}_{\mathbf{u}'})^{-1}\bar{\mathbf{m}}_{\mathbf{u}'} = K'^{1/2}_{\mathbf{u}'\mathbf{u}'}(I - \tilde{S}_{\mathbf{u}'})^{-1}(I - \bar{S}_{\mathbf{u}'})^{-\top}(I - \bar{S}_{\mathbf{u}'})^{\top}\bar{\mathbf{m}}_{\mathbf{u}'}.$$

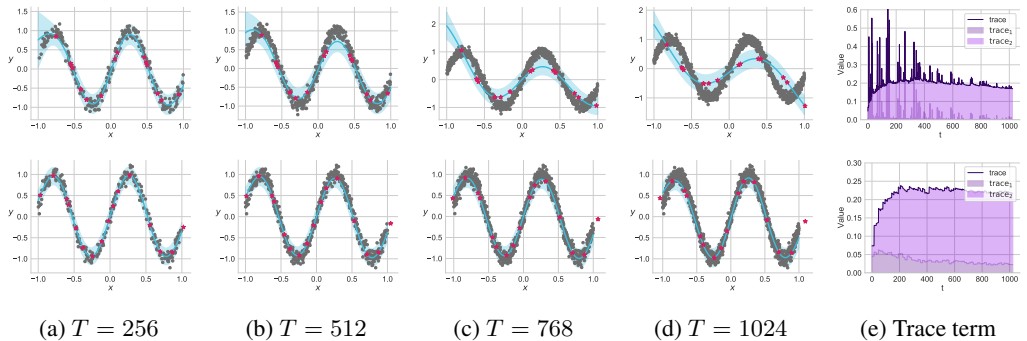

(a) $T = 256$    (b) $T = 512$    (c) $T = 768$    (d) $T = 1024$    (e) Trace term

Figure 6: O-SGPR ($p = 16$) learning on i.i.d observations without re-sampling the inducing points from a noisy sine function. **Top row:** As the number of data points increases for a batch size of 1, the model progressively underfits due to excess regularization. The trace term is entirely dominated by $\text{trace}_2$. **Bottom row:** For a larger batch size ($b = 16$), there is no under-fitting and $\text{trace}_1$ takes up more of the overall trace term.

From an engineering point of view, as we condition into an exact GP for most of our applications, we are able to use the pseudo-likelihood covariance alongside the cache of the pseudo-data covariance, caching a root decomposition of the matrix $(K_{\text{joint}} + \texttt{blkdiag}(\Sigma_{\hat{\mathbf{y}}}, \Sigma_{\mathbf{y}}))^{-1}$ via low-rank updates to a pre-existing root decomposition of $K_{\mathbf{uu}} + \Sigma_{\hat{\mathbf{y}}}$ (and its inverse), where $K_{\text{joint}} = k_\theta(\texttt{cat}(Z, X_{\text{batch}}), \texttt{cat}(Z, X_{\text{batch}}))$ [64, 39]. On performing several steps of conditioning (e.g. rollouts), we then use exact GP conditioning via low rank updates as implemented in GPyTorch (which uses the strategy of Jiang et al. [39] internally) after the first step.

### C.4 Incremental O-SGPR Tends to Underfit the Data

In this section we discuss a pathology of the ELBO derived in Bui et al. [9] that occurs when O-SGPR is updated on very small batches on new data (e.g. 1 new observation). Tellingly, Bui et al. [9] only considered tasks with large batch sizes (around 100 new observations per batch in each task).

Bui et al. [9] propose a "collapsed" evidence lower bound to train their O-SGPR model for each new batch of data. It is written as:

$$
\begin{aligned}
\mathcal{F}_{\text{OSGPR}}(\theta, \mathbf{u}) = {}& \log \mathcal{N}(\texttt{cat}(\hat{\mathbf{y}}, \mathbf{y}) | \mathbf{0}, Q_{\text{joint}} + \texttt{blkdiag}(\Sigma_{\hat{\mathbf{y}}}, \Sigma_{\mathbf{y}})) \\
& - \frac{1}{2}(\text{trace}_1 + \text{trace}_2) + \text{constants}, \\
\text{trace}_1 = {}& \sigma^{-2}\text{Tr}(K_{\mathbf{vv}} - K_{\mathbf{vu}}K_{\mathbf{uu}}^{-1}K_{\mathbf{uv}}), \\
\text{trace}_2 = {}& \text{Tr}((S_{\mathbf{u}'}^{-1} - K'^{-1}_{\mathbf{u}'\mathbf{u}'})(K_{\mathbf{u}'\mathbf{u}'} - K_{\mathbf{u}'\mathbf{u}}K_{\mathbf{uu}}^{-1}K_{\mathbf{uu}'})),
\end{aligned}
\tag{A.8}
$$

where $Q_{\text{joint}} = [K_{\mathbf{uu}}, K_{\mathbf{uv}}]^\top K_{\mathbf{uu}}^{-1}[K_{\mathbf{uu}}, K_{\mathbf{uv}}]$, and constants is composed of terms that do not depend on $\theta$ or $Z$. In a close parallel to the observations of Titsias [73] (e.g. Eq. A.1), we see that the O-SGPR objective is composed of a likelihood term and a trace term (written as $\text{trace}_1 + \text{trace}_2$), the latter acting as a regularizer [73, 3]. The first part of the trace term, $\text{trace}_1$, has the same interpretation as the trace term in the batch setting – it is minimized when the Nÿstrom approximation of the kernel matrix at $X_{\text{batch}}$ is exact (i.e. $K_{\mathbf{vv}} = K_{\mathbf{vu}}K_{\mathbf{uu}}^{-1}K_{\mathbf{uv}}$). The second trace term, $\text{trace}_2$, is minimized when $Z' = Z$, so it regularizes the new inducing point locations to be close to the old locations. If the batch size, $b$, is much less than $p$ then the trace term is dominated by $\text{trace}_2$, which is after all a sum over $p$ terms, compared to $\text{trace}_1$ which is a sum over $b$ terms. Since the loss encourages the model to keep $Z'$ close to $Z$ to minimize $\text{trace}_2$, the model can simply increase $\sigma^2$ to also decrease $\text{trace}_1$ to explain new observations by under-fitting. An analogous problem for small batch sizes was described in the Appendix of Stanton et al. [70] for the un-collapsed bound (e.g. the training procedure of an O-SVGP model) of Bui et al. [9], necessitating Stanton et al. [70] to propose a variant that down-weights the prior terms in the objective. The cost of down-weighting is an increased tendency towards over-fitting as well as an additional hyper-parameter, both of which were observed by Stanton et al. [70].

**Algorithm 2** LTS with SVGP

---

**Input:** Observed data $\mathcal{D} = (X, \mathbf{y})$; SVGP model, $\mathcal{M}$; candidate set generation utility, $C_{\text{gen}}()$, rollout steps, $T$, parallel path parameter $l$, top $k$ parameter, $q$ batch size.
**Output:** Candidates for evaluation $\bar{\mathbf{X}}_{\text{end}}$.

---

1. Generate initial candidate set, $X_1 = C_{\text{gen}}()$.
2. Compute posterior over candidate set, drawing a posterior sample: $y_1 \sim p(y|X_1, \mathcal{M})$.
3. Sort $y_1$ and keep top $l$ samples $\tilde{y}_0$ and corresponding candidates, $\tilde{X}_1$.
4. Generate $\mathcal{M}_1 \leftarrow \text{OVC}(M, (\tilde{X}_i, \tilde{y}_i).\text{unsqueeze}(-1))$ via Algorithm 1. $\mathcal{M}_1$ is a batch of $l$ models each conditioned on a single data point.
**for** t in 2:T **do**
    5. Generate initial candidate set, $X_t = C_{\text{gen}}()$.
    6. Compute posterior over candidate set, drawing a posterior sample: $y_t \sim p(y|X_t, \mathcal{M}_{t-1})$.
    7. Sort $y_t$ and keep top $l$ samples $\tilde{y}_t$ and corresponding candidates, $\tilde{X}_t$.
    8. Generate $\mathcal{M}_t \leftarrow \text{OVC}(\mathcal{M}_{t-1}, (\tilde{X}_t, \tilde{y}_t))$ using Algorithm 1.
**end for**
9. Generate $\mathcal{M}_{\text{end}} \leftarrow \text{OVC}(\mathcal{M}, (\tilde{X}_i, \tilde{y}_i)_{i=1}^T)$ using Algorithm 1.
Generate final candidate set, $X_{\text{end}} = C_{\text{gen}}()$.
10. Compute posterior over candidate set, drawing a posterior sample: $y_{\text{end}} \sim p(y|X_{\text{end}}, \mathcal{M})$.
11. Sort $y_{\text{end}}$ and return top $q$ corresponding candidates, $\bar{\mathbf{X}}_{\text{end}}$.

---

In Figure 6, we empirically demonstrate the pathology of the O-SGPR bound with small batch sizes in the i.i.d. setting. In the top row, we add $b = 1$ data point at a time while continuing to re-train. Although the O-SGPR model originally fits the data well at $T = 256$, it progressively begins underfitting, which becomes more and more noticeable, especially by $T = 1024$. By comparison, a larger batch size, $b = p = 16$, prevents any under-fitting from occurring. In the far right panel, we see the two terms in the trace component; in the small batch setting, the inducing trace term dominates the total trace. The effect is mediated by a larger batch size, as shown in the bottom right panel.

Informally, if the number of old inducing points is much greater than the number of new observations, then O-SGPR will focus on replicating the old variational distribution. Note that either aggregating multiple batches for each update or conditioning O-SGPR into an exact GP remedies the issue.

### C.5 Look-Ahead Thompson Sampling

**Thompson sampling in continuous domains**: in the context of black-box optimization, the action space is simply the input space, $\mathcal{X}$, since we are deciding which input $\mathbf{x} \in \mathcal{X}$ we will query next. Thompson sampling draws the next query point from the Bayes-optimal distribution over the possible choices, $\mathbf{x}^* \sim p_{\text{TS}}(\mathbf{x})$, where

$$p_{\text{TS}}(\mathbf{x}) \propto \int \mathbb{1}\{f(\mathbf{x}) = \sup_{\mathbf{x}' \in \mathcal{X}} f(\mathbf{x}')\} p(f|\mathcal{D}) df. \tag{A.9}$$

When $\mathcal{X}$ is a continuous domain, as is usually the case, we replace the $\sup_{\mathbf{x}' \in \mathcal{X}}$ with a $\max_{\mathbf{x}' \in X_{\text{cand}}}$, where $X_{\text{cand}} \subset \mathcal{X}$. As the name suggests, rather than attempting to evaluate the integral in Eq. (A.9), Thompson sampling instead draws samples $f_i \sim p(f|\mathcal{D})$, $i \in \{1, \ldots, q\}$ and take $\mathbf{x}_i^* = \text{argmax}_{\mathbf{x}' \in X_{\text{cand}}} f_i(\mathbf{x}')$.

**The look-ahead case**: If we were able to evaluate each $f_i$ on every point $\mathbf{x} \in \mathcal{X}$ and compute $\sup_{\mathbf{x}' \in \mathcal{X}} f_i(\mathbf{x}')$ exactly, there would be no benefit to multiple rounds of Thompson sampling. However, as we noted above, typically we rely on a $\max$ over a discrete set $X_{\text{cand}}$, typically obtained from a (quasi-)Monte Carlo method (e.g. Sobol sequences) to cover $\mathcal{X}$, and the number of candidate points is restricted by compute and memory. Therefore we can do multiple rounds of Thompson sampling to try to refine the estimate of $\mathbf{x}_i^*$ by evaluating $\text{argmax}_{\mathbf{x}' \in X_j} f_i(\mathbf{x}')$ for a sequence of candidate sets $X_j$, $j \in \{0, \ldots, h\}$. The key challenge with GPs is to ensure that $f_i$ is consistent across the sequence of candidate sets, which we accomplish by drawing $f_i(\mathbf{x}') \sim p(f|\mathcal{D} \cup \{(\mathbf{x}_{j-1}^*, f_i(\mathbf{x}_{j-1}^*))\}$ for $\mathbf{x}' \in X_j$ and $j > 0$. The result is again $\mathbf{x}_i^* = \max_{\mathbf{x}' \in X_{\text{cand}}} f_i(\mathbf{x}')$, but now $X_{\text{cand}} = \bigcup_j X_j$.

In Algorithm 2, we describe how OVC is used within LTSs as an example of its usecase. Here, of course, we are only performing Thompson sampling over a discrete set of values and so do not end up needing to use gradient based acquisitions. Specifically, we continue using OVC (or really after $T = 1$, exact GP conditioning with low-rank updates [39] as the GP is now exact), to condition our model on each step's fantasy responses $\tilde{y}_t$ and the observations $\tilde{X}_{\text{batch}}$.

# D   Further Experimental Details and Results

## D.1   Updating O-SGPR Inducing Points

We illustrate the efficacy of this choice of new inducing points in Figure 7 using the same time series data as in Stanton et al. [70] originally from `https://raw.githubusercontent.com/trungngv/cogp/master/data/fx/fx2007-processed.csv` (that repo uses BSD License). Re-sampling the old inducing points is shown in the top row, and tends to first perform well, but then begins to catastrophically forget by $t = 40$ and dramatically so by $t = 60$, as all of the inducing points have moved over to the right. By comparison, our approach of iteratively running a pivoted cholesky on the current inducing points and the new data point, prevents catastrophic forgetting, while also enabling the model to learn on the new data stream.

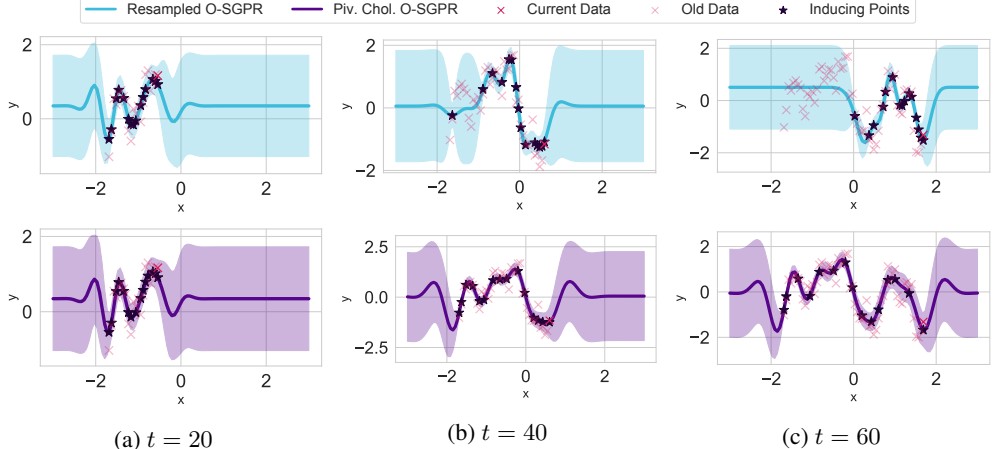

(a) $t = 20$    (b) $t = 40$    (c) $t = 60$

Figure 7: Online SVGP modelling a time series **Top row:** inducing points are updated by replacing an inducing point with the new location as in Bui et al. [9]. **Bottom row:** Inducing points are updated by re-running a pivoted cholesky on both the new data point and the current inducing points. The recursive refitting procedure of the pivoted cholesky placement remedies the catastrophic overfitting and forgetting of merely resampling the inducing points.

In Figure 8, we illustrate the effect of Laplace approximations during a rollout following the streaming classification example of Bui et al. [9] as we use OVC. We first trained a SVGP model with 25 inducing points on 100 data points, as shown in the first rows, then performed three steps of rollouts each with 100 data points each as we observe progressively more and more of the dataset. In the middle row, we show the predicted probability on a held-out test set; as we observe more data, the predictions become more and more confident throughout the entire region, and are un-confident in the regions where we have not observed any data. This effect is similarly observed by the predictive variances, which are high in regions where we have not seen any data, but decay as we observe each region successively. Data from `https://github.com/thangbui/streaming_sparse_gp/tree/master/data` (Apache 2.0 License).

## D.2   Experimental and Data Details

Unless otherwise specified, all data is simulated. The code primarily relies on PyTorch [62] (MIT License), BoTorch [2] (MIT License), GPyTorch [25] (MIT License). All GPs (variational and exact) used a constant mean, scaled Matern-$5/2$ kernels with ARD with lengthscale priors of `Gamma`$(3, 6)$ and outputscale priors of `Gamma`$(2, 0.15)$, which are the current BoTorch defaults for single task GPs.

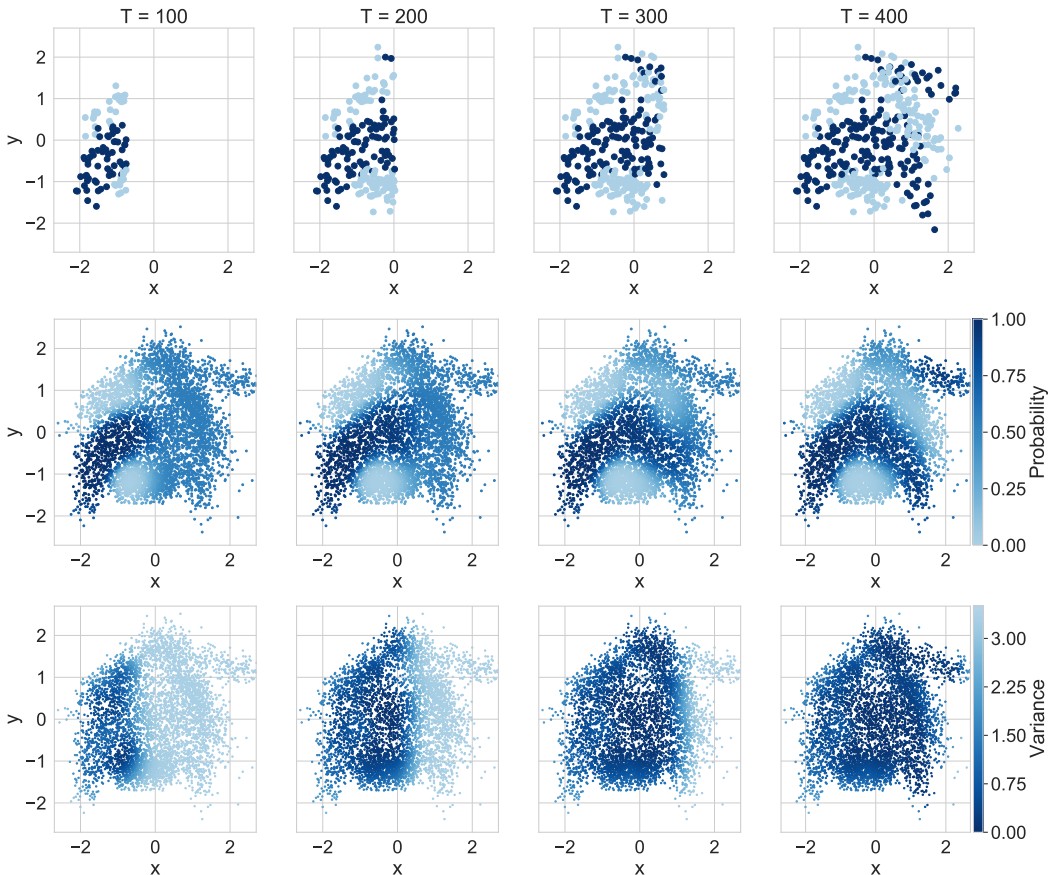

Figure 8: **Top row:** Data from the bananas dataset arriving in a non-i.i.d fashion in four sucessive batches. **Middle row:** Predictive probabilities of a SVGP with OVC to rollout conditional on these batches. Even in the non-Gaussian setting, OVC is able to adapt to new data without catastrophic forgetting. **Bottom row:** Variance of the latent function during the OVC rollout. The variances decay as we observe new data.

All variational GPs used GPyTorch's whitened variational strategy. Unless otherwise specified, we normalized all inputs to $[0, 1]^d$ and standardized outputs to have zero mean and standard deviation one during the model fitting stage. We trained all models to convergence with an exponential moving average stopping rule using Adam with a learning rate of $0.1$. We re-fit each model independently at each iteration. When plotting the median, we plot the $95\%$ confidence interval around it following `https://www-users.york.ac.uk/~mb55/intro/cicent.htm`. Unless otherwise specified, all acquisitions were optimized with multi-start L-BFGS-B with 10 random restarts and 512 samples for initialization for up to 200 iterations with a batch limit of 5 following Balandat et al. [2].

### Understanding Experiments

All understanding experiments, e.g. Figures 1, 6, 7, 8 were run on CPUs with Intel i5 processors. Computational costs were negligible to the cost of writing this paper.

**Figure 1:** We fit each model using the ELBO. The Gaussian function is $f(x) = \sin(2|x| + x^2/2)$ with $n = 100$ and $n_{\text{test}} = 25$.

For the GPCV model, we follow the model definition of Wilson and Ghahramani [82] and parameterize the scale of the Gaussian as a linear softplus transform, but we implemented a variational version, rather than the Laplace or MCMC implementations that are considered in the original paper. The data itself is a forward simulation of the well known SABR volatility model [30] with parameters $F_0 = 10$, $V_0 = 0.2$, $\mu = 0.2$, $\alpha = 1.5$, $\beta = 0.9$, $\rho = -0.2$. We model the scaled log returns and

plot volatility rather than the latent function. We use $n = 250$ and $n_{test} = 150$, so that $T = 400$; we standardize the inputs. Here, to perform Laplace approximations, we used PyTorch's higher order AD software as deriving the gradients and Hessians would be tedious.

**Knowledge Gradient on Branin:** We used the Branin test function as implemented in BoTorch [2] with $n = 50$ and 25 inducing points, 8 fantasies per data point, 250 candidate points and a grid of size $15 \times 15$. These were run on a single Nvidia Titan 24GB RTX. Computation took several minutes.

**Incremental Learning on Protein:** We followed the experimental protocol of Stanton et al. [70] but substituted in Matern-$5/2$ kernels with ARD instead of linear projections. The data comes from Dua and Graff [19]. The experiment is run over 10 random seeds and we show the mean and two standard deviations of the mean. Computation took several hours per trial.

**Batch Knowledge Gradient Experiments**

qEI and qNEI optimization used quasi Monte Carlo (QMC) integration with 256 random samples, while qKG optimization used QMC integration with 64 random samples (BoTorch defaults).

**Hartmann6:** Data comes from the Hartmann6 test function from `https://github.com/pytorch/botorch/blob/master/botorch/test_functions/synthetic.py` and the experiment is inspired by `https://botorch.org/tutorials/closed_loop_botorch_only`. These were run on CPUs on an internal cluster. Computation took several hours per trial. We used and 1000 randomly sampled candidate points to estimate the knowledge gradient.

**Laser:** Comparison scripts are from `https://github.com/ermongroup/bayes-opt/`. No license was provided. These were run on CPUs on an internal cluster. Computation took several hours per trial.

**Preference Learning:** Function is inspired by `https://botorch.org/tutorials/preference_bo`; we used noise of $\sigma = 0.1$ to make the function more difficult. The Laplace implementation comes from `https://github.com/pytorch/botorch/blob/master/botorch/models/pairwise_gp.py`. These were run on CPUs on an internal cluster. Computation took several hours per trial. qNEI optimization used QMC integration with 128 random samples, while qKG optimization used QMC integration with 64 random samples. Here, we used 3 random restarts and 128 raw samples for acquisition function optimization.

**Active Learning Experiments**

**Malaria:** Data is originally from Weiss et al. [81] under a creative commons 3 license, `https://malariaatlas.org/malaria-burden-data-download/#FAQ`. From the reference, the data is modelling predictions off of survey data and thus not human responses. For all models, we used Matern-$1/2$ kernels due to the lower smoothness and fixed noise models as variance is known. Comparison is to WISKI [70], with their code `https://github.com/wjmaddox/online_gp` which uses Apache License 2.0. These were run on a combination of Nvidia 32GB V100s and 48GB RTXes on an internal cluster. Here, we used 4 random restarts with 64 base samples to optimize the aquisition. Computation took several hours per trial.

**Hotspot Modelling:** Simulated data and comparison data is from `https://github.com/disarm-platform/adaptive_sampling_simulation_r_functions`. No license was provided for either. Our trials were run on AMD 32GB Mi50 GPUs on an internal cluster. Computation took close to eight hours per trial. We used tempering with $\beta = 0.1$ [38] and Matern-$3/2$ kernels for these models following the kriging setup in Andrade-Pacheco et al. [1]. Overall, we used 16 inner and outer samples for the entropy search objective, enumerating over all remaining test points to select a new point to query.

We note that the model fitting procedure of Andrade-Pacheco et al. [1] seems to possibly encourage test-set leakage as they seemingly use a random forest trained on all of the data, rather than on simply the first $n$ observations. See the third from final paragraph in their description of spatial methods in that section. We do not follow this as we do not use any random forests.

**TurBO Experiments**

**Rover:** We use the opensource Turbo implementation from `https://botorch.org/tutorials/turbo_1` and the rover function setup code from `https://github.com/zi-w/Ensemble-Bayesian-Optimization`. Both are licensed under the MIT License. These were run on Nvidia $24GB$ RTXes on an eight GPU server. We repeated experiments 24 times. See the wall-clock time panels for estimates of computational budgets (about an hour). As not all trials reached exactly $40,000$ iterations (due to cholesky decomposition errors, memory errors, and TurBO not restarting after 190 steps), we assume that the maximum achieved value was the best evaluation out to 200 steps to mimic the performance that one would see if using the method in practice. Early failures were only an issue for the exact GPs and then due to numerical instability, actually inspiring Figure 12a. As this truncation wreaked a bit of havoc with our timings, we only report the first 150 step timings in the main text (170 for Figure 13).

We used $500$ data points and the same model classes for the conditioning experiment (Figure 12a). Error bars are two standard deviations of the mean over the 10 paths used.

For the global models experiment (Figure 12c), we used a combination strategy that first used half the batch with Thompson sampling (TS) to select the points and then used qGIBBON [54] to select the other half of the batch by setting the TS half of the batch as pending points. Performance using qGIBBON alone was about twice as slow and was slightly worse due to the lack of exploration that TS provides. This strategy also enforces that qGIBBON's implementation actually uses fantasization and OVC, which would not natively have been the case without using the pending points. For the timings, we report the first 106 steps over 8 seeds.

**MuJoCo:** We use the codebase of Wang et al. [78] including their patched TurBO implementation with an optimization loop. This codebase is available from `https://github.com/facebookresearch/LaMCTS/tree/master/LA-MCTS` with a Creative Commons 4.0 License with the included, modified TurBO implementation following a non-commercial license. The MuJoCo experiments use mujoco-py (`https://github.com/openai/mujoco-py`, MIT License) and an institutional license key for MuJoCo itself [75]. These were run on a combination of Nvidia 32GB V100s and 48GB RTXes on an internal cluster. We repeated these experiments over 10 trials and computation took close to 14 hours for hopper (where one trial failed to reach 4000 samples for all methods but exact LTS), and several hours for swimmer.

On hopper, we struggled with wide variation in model fits, so we changed the regularization strategy on the base GP models to account for the high dimensional feature space. Inspired by Eriksson and Jankowiak [21], we continued using ARD Matern-5/2 kernels but placed `HalfCauchy($\tau$)` priors on the inverse lengthscales and then placed a `HalfCauchy(1.0)` prior on $\tau$ itself. Rather than using MCMC as Eriksson and Jankowiak [21] did, we used MAP to estimate both the lengthscales and $\tau$.

## D.3 Bayesian Optimization with the Knowledge Gradient

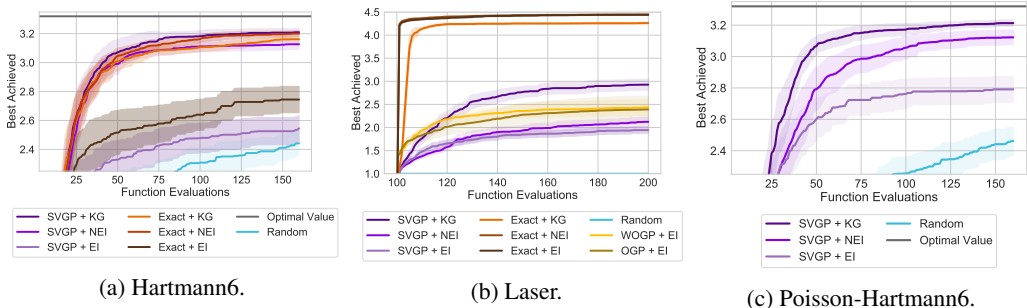

Figure 9: **(a)** Comparison to a broader suite of methods on Hartmann6, 1 constraint. **(b)** Comparison includng BoTorch exact GPs and their acquisitions on the free electron laser problem. Comparing to the results in Figure 3 of McIntire et al. [51], these exact GPs vastly outperform their implementation, presumably due to advances in acquisition function optimization. **(c)** Hartmann6 test problem with count responses (Poisson likelihood). Only approximate inference can be used here, and qKG vastly outperforms qNEI.

In Figure 9, we present the results for a wider set of acquisition functions using the one-shot knowledge gradient on three test functions. These results complement Figure 3 and only include these additional methods. Overall, qKG (with either an exact GP or a SVGP) generally performs best, followed by qNEI and then qEI.

In Figure 9a, we include results on the constrained Hartmann6 problem with random baselines as well as exact and SVGPs with expected improvement (EI). Exact and SVGPs with EI are signficantly outperformed by the other, more advanced acquisitions, but do outperform a random baseline.

Figure 10: Best achieved values on Hartmann-6 for batch size, $q = 3$ for both NEI and KG for exact and SVGPs.

| Noise Level | Acqf | Method | Best Value |
|---|---|---|---|
| 0.1 | NEI | Exact | 3.19 (0.05) |
| 0.1 | NEI | SVGP | 3.08 (0.06) |
| 0.1 | KG | Exact | 3.17 (0.03) |
| 0.1 | KG | SVGP | 3.13 (0.04) |
| 0.5 | NEI | Exact | 3.20 (0.03) |
| 0.5 | NEI | SVGP | 3.12 (0.06) |
| 0.5 | KG | Exact | 3.15 (0.03) |
| 0.5 | KG | SVGP | 3.21 (0.05) |

In Figure 9b, we also show the results of exact GPs as well as an online GP [OGP, 16] with EI using the implementation of McIntire et al. [51]. Interestingly, all of the exact GPs significantly outperform their variational counterparts. This is quite surprising in some sense, as the true simulator is a weighted OGP with fixed hyper-parameters, and this method (WOGP + EI) performs much worse. Note that the random baseline makes no progress.

Finally, in Figure 9c, we display the results on constrained Hartmann-6 with Poisson observations, where each method outperforms random querying, but as expected SVGP + qEI, performs worse than qNEI and qKG.

**Batch Size and Noise Level Ablation:** In Tables 10 and 11, we display the final optimization results after 150 function evaluations on the Hartmann-6 test problem for varying levels of noise and for each acquisition. These results are over 20 trials and we display the mean maximum achieved value.

Overall, KG tends to outperform NEI at both low and noise levels, with both exact and SVGP models performing very similarly overall with the SVGPs getting a slight edge in the high noise setting. Furthermore, larger batch sizes tend to perform slightly better as the mean maximum achieved tends to have lower variation. Finally, higher noise levels tend to be somewhat harder to optimize as expected.

As we add more Gaussian noise into the function, we might a priori expect that qNEI should outperform qKG given the class of models. However, all things being held equal, a model that is more robust to the observed noise should tend to perform better, particularly if we are using more than just its mean and variance. Thus, SVGP models, by virtue of having more parameters to tune, tend to be more robust to the observed noise than the exact GPs.

### D.4 Ablations on Rover

Figure 11: Best achieved values on Hartmann-6 for batch size, $q = 1$ for both NEI and KG for exact and SVGPs.

| Noise Level | Acqf | Method | Best Value |
|---|---|---|---|
| 0.1 | NEI | Exact | 3.14 (0.13) |
| 0.1 | NEI | SVGP | 3.10 (0.10) |
| 0.1 | KG | Exact | 3.20 (0.02) |
| 0.1 | KG | SVGP | 3.18 (0.03) |
| 0.5 | NEI | Exact | 3.10 (0.10) |
| 0.5 | NEI | SVGP | 3.10 (0.14) |
| 0.5 | KG | Exact | 3.12 (0.04) |
| 0.5 | KG | SVGP | 3.16 (0.05) |

**Effects of LTS:** To ablate the effects of rollouts and improved conditioning, we consider several step rollouts on the rover function as shown in Figure 12a. We find that performance is similar across depths. However, the conditioning of the resulting training data covariance is vastly improved when using OVC, as shown in Figure 12b. Taking these two results together, we see that in most cases, a tree depth of 4 should be enough to gain the improvements from rollouts without increasing the conditioning of the system too much.

**Global models and SGPR:** To further demonstrate time efficiency of using OVC in the context of even global models, we perform large batch BO with the recently introduced qGIB-

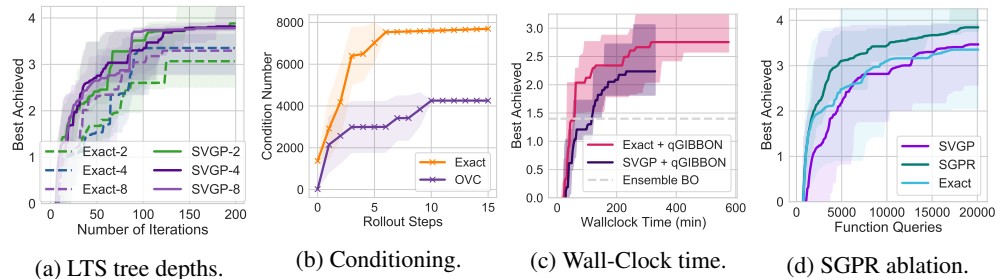

(a) LTS tree depths.  (b) Conditioning.  (c) Wall-Clock time.  (d) SGPR ablation.

Figure 12: **(a)** Conditioning of exact and OVC conditioning across LTS tree depths on rover. **(b)** Performance of different rollout depths is similar, as the bigger gains are conditioning. **(c)** Time efficiency using global models. SVGPs are still a strong baseline. **(d)** Comparison with SGPR using pivoted cholesky initializaiton on rover. SGPR is competitive.

BON acquisition (a max value entropy search variant) but using half the batch with TS to enforce fantasization over the TS-acquired batch [54]. This strategy is significantly slower than TurBO; however, even in this setting using SVGPs is twice as fast, and achieves a similar result to the exact model, as shown in Figure 12c. Both are orders of magnitude faster than Ensemble BO, which uses batch max value entropy search and exact GPs with addtive kernels [80, 79]. Ensemble BO takes at least several days of compute time [22]. Our result here compares very favorably to the (un-timed) results using SVGPs as well as exact GPs with ARD kernels that Wang et al. [79] also compared to, as neither of those methods reached reward values $\geq 1$ on this problem, even after $35,000$ steps.

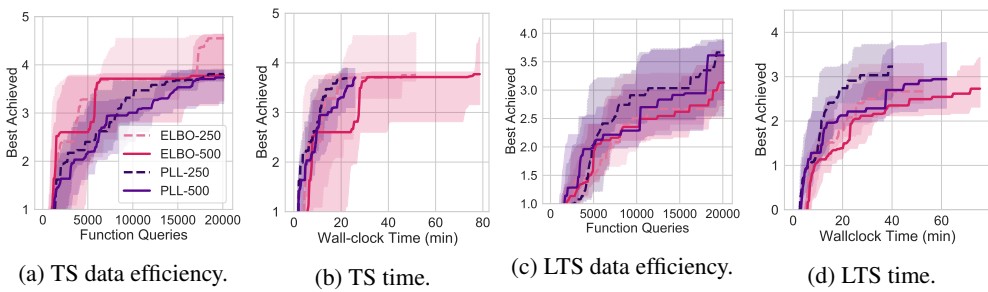

(a) TS data efficiency.  (b) TS time.  (c) LTS data efficiency.  (d) LTS time.

Figure 13: Inducing point ablation on rover **(a,b)** use Thompson sampling and **(c,d)** use rollouts.

**Number of Inducing Points and Training Loss:** Finally, in Figure 13 we ablate between training SVGPs with either $250$ or $500$ inducing points as well as the training loss, either the evidence lower bound (ELBO) or the predictive log likelihood (PLL), (see Appendix B.3 for further descriptions) on the $d = 60$ rover problem. We find that there is not a significant amount of difference between any of the four approaches whether using the ELBO or the PLL. In general, the PLL approaches are somewhat more quick to train (Figures 13b and 13b) in comparison to the ELBO models. They also tend to slightly outperform the ELBO-trained SVGPs when using LTS (Figure 13c) in terms of function efficiency, but perform similarly for standard Thompson sampling (Figure 13a). We leave a detailed benchmarking of these methods in the context of downstream tasks for future work.