# OpenReview forum: "Conditioning Sparse Variational Gaussian Processes for Online Decision-making"
_NeurIPS.cc/2021/Conference — NeurIPS 2021 Poster_

### Official Review · Reviewer_XKNN · 2021-07-14

**Rating:** 7
**Confidence:** 3

**Summary:**

Authors propose a novel method, named OVC for online variational conditioning of GPs under sparse approximations. Particularly, they are interested in the recursive update of the posterior GP distribution after receiving new data per time-step. The model can be used for both regression and classification tasks, bayesian optimisation and active learning as is shown in the experiments.

**Limitations And Societal Impact:**

Authors adequately addresses both limitations and societal impact of the work in the section A of the Appendix. Particularly, they explained very well the limitations of the method and the cases where it might not work very well.

**Main Review:**

**Originality and significance:**

Good paper. I enjoyed quite a lot while reading it. Particularly, I want to remark the following points that are positive to me:

- The idea of transforming the inducing points and their corresponding function instances into pseudo-data is interesting. It can be considered as some sort of data augmentation. Authors correctly identified the limitations of the approach presented in Bui et al. (2017).
- Figure 1 is great. It is not that easy to obtain such performance, even if the “smoothness” of the signal changes across time. How do you treat the kernel hyperparameters? In that particular case seems like between [-2,2] the length scale is around 1.0, and later between [2,6] it is smaller (0.5 for example). Is the model well fitted to the past data because you are using the cache thing right? Otherwise would be problematic if re-training.
- Experiments are well designed and prove the performance of the tool. I see clearly that the method outperforms Bui et al. (2017).
- Perhaps Bui et al. (2017) could work better with non-Gaussian likelihoods, but the limitations are overcame by OVC here. I like the Laplace approximation, but maybe if considering multivariate likelihood models, things could be more complicated..
- I found the “banana dataset” experiment in Figure 8 of the Appendix. What a pity not having included it in the main manuscript, because the results are pretty good.
- An extra positive point of this work is that you are able to update the model with one-single observation (in a filtering-like model) or with more than one data-point per time step (similarly as Bui et al. 2017). Great!

**Quality and clarity:**

 In particular, I want to remark the quality of Section 2, which demonstrates the wide knowledge of authors on the online GP topic and how the decisions taken are well-motivated later. Experiments are also thorough and very well presented. Results are convincing. To sign a flaw of the paper, I would say that section 3.1 could be clearer, but this is a little detail. I’m sure the paper will have an impact on the community.

**Questions and other comments:**

- L106: I think that this whitened parametrisation is not that well-known, I suggest a super-short explanation, mainly because later is re-used.
- L156: are the definitions of u’ and u in the correct order? It seems that u’ should refer to f(z’), instead of the written ones.
- The approximation of y_u in the point 1 of Algorithm 1) makes sense to me, but maybe is not that well explained. Do you do it to use the “fixed” variational distribution S_u, right? Otherwise, would be useless at the next time-step t.
- The cache terms a and RR look like they are gonna be critical in the paper at the beginning, but later they are not commented anymore. The task of adding new rows and columns to Kvv (L82) in an online fashion seems like a contribution of other works, is that the reason why authors do not claim it?
- Algorithm 1 would be even better having a recursive nature, like do this every time step t. At point (4) one may get lost and do not know how to use the model again at t=t+1.
- L192: How is the number of inducing points treated? Do authors augment them as long as new data arrives? Which is the criteria?
- Could you add some extra details about the treatment of hyperparameters + kernel models? Are you assuming stationary functions or it may change?

**References:**

- Bui et al. (2017). “Streaming sparse Gaussian process approximations”. NIPS 2017.




**Time Spent Reviewing:**

3,5

---

> ### Author Response · Authors · 2021-08-10
> **Thanks for the positive feedback**
>
> Thank you for the very detailed and uplifting review. We provide our responses below:
>
> - L106: Indeed, this parameterization is a bit different from the standard parameterization of sparse variational Gaussian processes and seems to be closely related to so-called dual parameterizations of exponential families. We will add a comment that in the sparse Gaussian process regression (SGPR, Titsias ‘09) case, it seems to be the optimal projection of the n observed data points into the space of the m inducing points.
> - L156: Yes, good catch.
> - Alg 1, $y_u$: Yes, $y_u$ is re-computed at every iteration and is used to correct the update to both $m_u$ and $S_u$. We will clarify this.
> - Alg 1, recursion: We chose to write Algorithm 1 as it is because of its usefulness in depicting the pseudo-data formulation that we use (e.g. returning an exact GP rather than an SVGP). The recursive formulation is given in Eqs 7 and 8. We can add a secondary algorithm box in the supplementary using the recursive formulation.
> - Caches, L82: Indeed, the problem of online updates to K_{vv} has already been studied. We choose to highlight them due to their usefulness in sample based Bayesian optimization; with cache computation, posterior sampling is straightforward.
> - Inducing points: In general, we choose a fixed number of inducing points. For example, in the rover experiments, we choose 500 fixed inducing points and update their locations as we both train and add new data.
> - Kernel hyperparameters: For SVGP models, we train the kernel hyper-parameters, the inducing locations, and the variational distribution’s parameters through the evidence lower bound (given in Eq. A.2). This approach seems to be quite common throughout the variational GP literature, see the experimental description in Hensman et al, UAI '13 ("Gaussian Processes for Big Data") and Shi et al, AISTATS, '20 ("Sparse Orthogonal Variational Inference for Gaussian Process").
> - Yes, the strength of the fit will certainly play a crucial role in the success of the rollout steps.
>
> edited for formatting.

---

> > ### Comment · Reviewer_XKNN · 2021-09-01
> > **After Authors' Rebuttal**
> >
> > Thanks to the authors for their response and the effort dedicated to answer the reviews. The response addressed the main points that I suggested. As I said, I liked particularly the idea of the paper, which I consider novel and the quality of experimental results. I also think that that the main point of improvement could be the end of Section 2 and a bit more clearer Section 3.
> >
> > Some details raised by other reviewers make me difficult to raise my score, particularly the technical ones on the mistake in Eq. 3, the parametrization based on exponential families or the partial isolation of the Bayesian opt. setup in the story of the paper. So I will keep as it is, with a lower confidence. But I want to remark once again that I liked the paper. Additionally, if allowed in future updates, Illustrative Figures as Fig. 1. in the experiments with streaming data are always welcomed in this sort of works, and very much appreciated by the readers.

---

### Official Review · Reviewer_pE77 · 2021-07-14

**Rating:** 6
**Confidence:** 4

**Summary:**

The streaming sparse GP (named O-SGPR in this paper) is a variational GP approximation that supports updating the variational posteriors and hyper-parameters online. This paper reinterprets O-SGPR as compressing the past observations into a small set of "pseudo-data". Then updating the posterior based on new observations can be simply treated as a standard GP inference based on the new observation and the pseudo-data. This paper adopts the proposed Online Variational Conditioning (OVC) over Bayesian optimization and active learning, where inferring the updated belief from new observations is important to quantify the contained information. This paper is well-written and conducts extensive experiments to verify the proposed approach.

**Limitations And Societal Impact:**

To further improve the paper, I think these are potential directions,
1) A clear discussion on the contribution compared to O-SGPR and a standard SVGP.
2) A clear model description on

 (a) how the variational distributions are obtained,

 (b) how the hyperparameters are chosen,

 (c) what is the training workflow,

This paper does not lead to negative societal impacts.

**Main Review:**

## Method
### Novelty and Significance
Enabling variational GPs for online posterior inference is an important problem. The streaming sparse GP (named O-SGPR in this paper) is a landmark approach that enables updating both the posterior belief and the hyper-parameters for variational GPs. This paper reinterprets O-SGPR in terms of performing GP inference over the new observations and "pseudo data", which I think is a novel contribution.

### Relations to O-SGPR
As discussed in the paper, OVC is based on a new interpretation of the O-SGPR. However, I think it deserves more discussions on their differences. Based on my understanding, O-SGPR updates the variational posterior by maximizing the ELBO, which prevents it from being directly plugged into the Bayesian optimization problems. In contrast, the OVC directly obtains the posterior by the "exact GP formulation", enabling the explicit reasoning in Bayesian optimizations. However, OVC sacrifices the general applicability towards non-Gaussian likelihoods and turns to Laplacian approximations.

I think this difference is the major contribution in this paper, given the existence of O-SGPR. And it is the key for being applied in Bayesian optimizations. But it is not obvious from reading the paper.

###  Relations to standard SVGP
Imagine that we have the inducing points $Z_{t-1}$ and the variational posterior $ \phi_{t-1}(u') = \mathcal{N} (\mu, S) $, then the current posterior is a Gaussian process
$$
GP_{t-1} = \mathcal{GP}( k(\cdot, Z) k(Z, Z)^{-1} \mu, k(\cdot, \cdot) - k(\cdot, Z) k(Z, Z)^{-1} k(Z, \cdot)  ),
$$
Given any new observations, we can also infer the new posterior belief under this $GP_{t-1}$.

Is OVC equivalent to this approach ? If not, what is the advantage of OVC against it ? I noticed that Line147-150 might be relevant, but the two shortcomings listed there apply to OVC as well in my perspective.


## Clarity
This paper is well written in general. Here are some minor comments,
1) Line 158: define $p_t, q_t$ before using them.
2)  Inducing Point Selection: do you select over "[new_obs, pseudo_data]", or  keep all "pseudo_data" and only select from the "new_obs" ?
3) Figure 2: The captions for b,c,d  are not clear. It is not clear to me what is the difference between (c) and (d).
4) Line 277: the notation $x$ is used in both $a(x; D)$ and $dx$.

## Experiments
1) Do you keep a fixed budget for the number of inducing points, or add a fixed number of inducing points per iteration ?
2) What is the impact of the number of inducing points ? For example, in Figure 3(a) and Figure 5, it is not clear to me how many inducing points you used compared to the Exact GP.
3) Do you update the kernel hyperparameters along training ?

**Time Spent Reviewing:**

4

---

> ### Author Response · Authors · 2021-08-10
> **Thanks for the supportive review**
>
> Thank you for the supportive review! We are happy that you found the paper to be clear and thoroughly evaluated, and we hope you can consider raising your score in light of our responses.
>
> **Relation to O-SGPR**
> Indeed, your understanding of O-SGPR is essentially correct, and we will further emphasize these distinctions in the camera ready. As O-SGPR updates the posterior implicitly it has qualitatively different performance than updating the variational posterior explicitly (as demonstrated when we applied it to profiling out the KG acquisition function in Figs 2b-d). However, by updating the posterior implicitly, O-SGPR is also limited to Gaussian observations and doesn’t have a native mechanism for updating in the case of non-Gaussian observations without a lot of re-training through the variational lower bound.
>
> **Connection to SVGP**
> Based off of our understanding of what is written, this GP is defined as a posterior distribution with responses given by $m$ the variational mean. As a thought experiment, one could continue adding new data to that model, by effectively stacking the true responses onto $m$ and the data inputs to Z, producing $\tilde Z$. However, this will become infeasible as we add a lot of new data points, because the kernel matrix $K(\tilde Z, \tilde Z)$ will grow (and thus the prior distribution becomes more and more expensive to compute (computation of its cholesky factor grows at O((M + n_{test})^3)). However, if we instead kept the inducing points constant and just re-computed the variational mean, that would require looking back at all past data and thus would also become prohibitively expensive.
>
> These two failure modes motivate our approach, OVC, which uses a recomputation of the inducing points to prune the size of the joint inducing input new test point matrix, while also optimally computing the update to the variational mean when possible.
>
> **Experiments**
> - Inducing points: We used a fixed budget of inducing points.
> - Fig 3a: We used min(N, 25) inducing points for all four trials in these experiments. At initialization, we often had fewer than 25 data points so we used N inducing points if we had N data points.
> - In Figure 5, we used 500 inducing points for the SVGPs. Please see our study of the effect of the number of inducing points in Appendix Fig 11 a-d. In general, we found that more inducing points is better for modelling; however, it can be more difficult to optimize a larger variational distribution.
> - Update kernel hypers: Yes, we trained the kernel hypers during training, see the next section of the rebuttal.
>
> **Model Description**
> In general, a Bayesian optimization loop consists of the steps of training the model and then using the trained model to optimize an acquisition function to acquire new data points, which are then added into the training data for the next model.
>
> In the first step, we first train the inducing points, variational distribution, and kernel hyper-parameters using the evidence lower bound given in Eq. A.2. As all components are differentiable, we use the Adam optimizer with a learning rate of 0.1 and optimize for 1000 steps or until the loss converges, whichever is shorter. To initialize the inducing points, we compute a pivoted cholesky factorization on the initial kernel on the training data (described in Section 3.2). The kernel hyper-parameters are initialized to GPytorch defaults (which sets all lengthscales to one), while the variational distribution is initialized to m = 0, S = I (again, GPyTorch defaults).
>
> **Clarity**
> - L158, L277: Thanks for the pointers; we will fix these typos.
> - Inducing Point selection: Yes, we select the new inducing points from the set: [Inducing points, new obs]. We will clarify.
> - Fig 2: This figure helps to distinguish between OVC (our approach) and O-SGPR. By “conditioning into O-SGPR” we mean following the approach of Bui et al, ‘17 (plus updating the inducing points) and on adding the new data points into the model, returning a sparse Gaussian process model (and thus predictive distribution) that has updated inducing locations and the specialized likelihood given by Eq. 6. By “conditioning into an exact GP”, we mean OVC (our approach) which combines the inducing locations and the new responses and uses the same likelihood. In this case, O-SGPR has a fixed inducing point budget and reduces to OVC if all inducing points and new data points are included in the model.

---

> > ### Comment · Reviewer_pE77 · 2021-08-17
> > **Response After Rebuttal**
> >
> > Thank the authors for the detailed responses. I have two more questions,
> >
> > 1, In Line 230, you mentioned that "the Hessian is psd and diagonal when the output-space distribution is an exponential family". However, I am skeptical whether this is true. For a softmax distribution, the logits are the natural parameters. In consequence, if I derived it correctly, the derived Hessian matrix would be `\diag(p) - p p^{\top}` when `p` is the class probabilities corresponding to the logits. Therefore, I don't think that the Hessian would always be diagonal. Am I wrong about this ?
> >
> > 2, For "Connection to SVGP", you mentioned the cubic computations for the naive SVGP approach. However, if I understand correctly, by storing $M$ OVC inducing points, the computational costs would also be $O( (M + n_{test})^3 )$. How could OVC improve over this ?
> >
> > And I am thinking of fixing the variational distribution $(m, S)$ as the "pseudo labels" after they are selected in a standard SVGP. So it does not need to be recomputed in each iteration. Will this cause a problem compared to OVC ?

---

> > > ### Author Response · Authors · 2021-08-18
> > > **Additional Comments**
> > >
> > > Thanks for the updated comments; we hope these responses can be updated in your final consideration.
> > >
> > > 1) Yes, this is a bit of imprecision, which we will clarify in the camera ready. The Hessian is diagonal if we restrict the setting to one-dimensional likelihoods (the critical aspect is really that each response is conditionally independent of the other responses given $f$). For softmax / categorical, there are several outputs per label and so the Hessian is no longer diagonal, but block-diagonal (your derivation is correct and produces dense blocks of C x C like you point out).
> > >
> > > 2a) We improve over this time complexity by using a pivoted cholesky decomposition, which reduces to $O((M + N_{test})(M^2))$ pre-computation and then $O(M^3)$ computation afterwards.
> > >
> > > 2b) Not quite, to use m, S as “pseudo-labels” we must approach the problem in a slightly different fashion to maintain the structure of conditional probability (e.g. the derivation up to Eq 3 in 3.1 of Bui et al, ‘17).
> > > Plugging in m S as the pseudo-likelihood would be a good initial attempt. However, even in the setting of their optimal values and for the inducing points as the original data points, then the joint distribution over both m and y_{new} is not the same as the joint distribution over y_{old} and y_{new} as produced from the GP model. It's only in the noiseless limit (\sigma^2 -> 0) that the two correspond.
> > >
> > > To recover the same probabilistic model, we (as Bui et al, '17 do as well) instead develop the pseudo-targets as the data in a given Bayesian inference setting with prior given by the GP prior over the inducing locations and posterior given by the variational posterior (with m, S as the parameters). This construction produces an inverse problem that we solve in Appendix C, so that the pseudo-targets and their associated covariance is given as the blue terms in Eq. 5.

---

### Official Review · Reviewer_DFYr · 2021-07-19

**Rating:** 5
**Confidence:** 4

**Summary:**

The authors are interested in the problem of online learning with applications to online Decision making. The focus is on (sparse variational) Gaussian Processes and Bayesian optimization.

The authors 1) propose an alternative formulation of the online learning algorithm of Bui et al, which allow to ‘preserve’ what has been learned from the previous data 2) propose a way to update the posterior when adding observations in the conditioning set for non Gaussian likelihood 3) Thoroughly evaluate the resulting algorithm on various BO tasks.


**Limitations And Societal Impact:**

Limitations: the authors could be more careful in discussing and assessing the limitations of the approximation introduced for the online learning problem before applying it to decision making.

negative societal impact : None

**Main Review:**


Originality and Quality:

The paper builds up on the work of Bui et al, where the aim is to update posterior processes when adding observations to the conditioning set.
The authors acknowledge mostly rederiving the method in an alternative fashion with the aim to connect it to Sparse Gaussian Process Regression (SVGP in the conjugate case by Titsias).

The approach consists in essence in explicitly representing the posterior via sites that are updated in an incremental fashion. The ‘unifying’ paper of Bui would be a welcome reference here. As a rederivation rather than introduction, I m not sure it is worth a new name. Also the terminology is not very enlightening. Computing the posterior is always conditioning on observations. online conditioning is not very precise not specific to this method (also see comment below).

The proposed incremental formulation is equivalent to Bui et al, and brings practical advantages when implementing. The particular applications of SVGP to BO is original (recent work on Thompson sampling in this particular combination would be worth mentioning :Vakili 2020).

I have a concern regarding the Laplace approximation. Each likelihood term is optimized independently of the ongoing posterior distribution. For a Bernoulli likelihood, the maximizer is at infinity and the hessian is ‘flat’ (being exponential family doesn't solve that). This problem is why the community has derived ways to compute sites in the context of the ongoing approximation  (khan et at 2017 for VI, or Bui et al 2017 for EP). This and other limitations of this heuristic should be discussed. The lack thereof does not make me confident or provide intuition that errors don’t accumulate.
An experiment focusing just on the quality of this approximation for online learning would help justify this particular choice.


Clarity:

The manuscript is fairly well written (see comments below for improvement suggestions)

Significance

Look ahead objectives are strong candidates for BO and deriving practical algorithms to update posterior processes approximately and cheaply is needed.
Though tackling an relevant problem, the contribution in the paper is a rewriting of an existing algorithm and I have concerns about the extension.

My rating of this paper reflects these mixed feelings, but I am ready to increase my score.


Remarks:

Eq 3: covariance looks wrong to me. you can’t subtract a covariance from a precision, the following derivations look fine though

‘Conditioning a GP into another GP’ is very informal and doesn’t help the reading. “conditioning into” is not a thing. I’d recommend using more standard formulations.

The S in acronym SVGP is used to denote both Sparse and Stochastic throughout the paper, which is confusing.

A couple of typos (the prime) in section 3.1, makes it hard to parse.


I have not thoroughly reviewed the experimental section and leave this to other reviewers.


**Time Spent Reviewing:**

5

---

> ### Author Response · Authors · 2021-08-10
> **Thanks for the thoughtful feedback**
>
> Thank you for your thoughtful review. We respond in detail to your questions and comments, and would like to make several clarifications. We hope you will consider raising your score in light of our responses and clarifications.
>
> **Connections with Bui et al, ‘17**
>
> Thank you for the pointers (including the references to [Bui, Yan, and Turner, ‘17] and [Vakilli et al, ‘20] which we will cite). We’d like to clarify our usage of the term “conditioning”. Our work focuses specifically on the following problem: given a posterior p(f | D) and some new data D’, how can we efficiently compute p(f | D, D’) without increasing our computation budget or looking back at D? There are efficient methods for exact GP regression (low rank updates, e.g. Chapter 5.2 of [Osborne, ‘10] or the procedure of [Jiang et al, ‘20]). If these same methods are applied to variational GPs, they require either adding new inducing points at locations in D’ (violating the constraint on compute) or they require variational inference on the joint dataset (violating our second constraint). Satisfying both requires that the variational distribution itself be updated, but not through standard variational inference for GPs.
>
> Although OVC is inspired by our rederivation of O-SGPR, it is distinct in the following ways:
>
>
> 1) O-SGPR, like its namesake SGPR, computes the optimal variational distribution implicitly at each timestep. A major contribution of our work is the explicit characterization of the implied updates to the variational distribution, allowing us to extend the method to SVGPs.
>
>
> 2) Bui et al [‘17] do use the term pseudo-data, but the usage is quite vague, because it is used to refer not just to the inducing point locations, but the inducing point function values, which are random variables in a GP model, and hence not data in any concrete sense. We reserve the term only for the result of projecting the variational distribution to $(Z, y_u)$. One consequence of this conceptual contribution is that we can easily transform an online variational GP into an equivalent exact GP, which is quite helpful for tasks like black-box optimization.
>
>
> 3) O-SGPR only describes how to efficiently compute the updated ELBO and predictive distributions. It makes no prescription for how the inducing point locations should be updates, which is an absolutely crucial detail for good performance. Our extension of the inducing point initialization scheme of Burt el al [‘19] is a key contribution that makes OVC a complete prescription for how online updates should be performed. Our update is significantly better than the resampling heuristic in Bui et al [‘17], which forgets old data, as we demonstrate in Fig 2a, Appendix C.4 (Figure 6), and in Appendix D.1 (Figure 7)).
>
>
> 4) O-SGPR closed form updates are only available for the Gaussian likelihood case in [Bui et al, ‘17] (this is the difference between their collapsed bound and their un-collapsed bound for fixed hyper-parameters), whereas we are able to extend their approach to non-Gaussian likelihoods via Laplace approximations without re-training (see below).
>
>
> 5) Finally, we push towards a unification of conditioning procedures for both variational Gaussian processes and exact Gaussian processes. For example, in Appendix C.3, we demonstrate that our approach is actually a generalization of the “exact” conditioning (e.g. posterior updating in response to new data) for the sparse Gaussian process regression model. This connection should help produce new methodologies that bridge the methodological gap between variational and exact Gaussian processes.
>
> **Laplace Approximation**
>
> *Practicality:* We point you to Figure 8 in the Appendix where we roll out three steps of the Laplace approximation on a binary classification problem, finding that the Laplace approximation pleasingly performs quite well in this setting, even in terms of predictive variances across the dataset (the bottom row there). In a Bayesian optimization setting, we also test out the effect of different numbers of rollout steps and their numerical conditioning in Appendix Figures 10 a and 10 b. These additionally demonstrate that rollouts of beyond 4-8 steps seem to have diminishing returns, even for Gaussian likelihoods (due to worse numerical conditioning). Finally, we mention that long rollouts and incorrect modelling can yield incorrect predictions in our discussion of our work’s limitations in Appendix A.
>
> *Theory:* Indeed, there is a slight bug, thanks for pointing this out. We noticed that Section 3.4 misstates the optimization problem solved by Laplace approximation, which we will correct. The optimal f is $\arg \max_f p(f | X, y) = \arg \max_f \log p(y | f) + f^\top K^{-1} f$, not $\arg \max_f \log p(y | f)$ (see Ch. 3 of Rasmussen & Williams, [‘06] specifically Equation 3.12). In other words, we do not optimize the function values independently of each other; the elements of f are optimized jointly due to the regularization term $f^\top K^{-1}f$. Thus, the un-regularized problem will be ill-defined for a Bernoulli likelihood: even under our parameterization f tends to infinity for y = 1 and to negative infinity for y = 0. However, the regularized problem is well-defined: for $k(x,x) =1$ and $y =1$, the stationary point for a single $f$ (according to WolfamAlpha) is at about $0.40.$ We regret the error and will correct it in the camera-ready.
>
> ***Remarks***
> - Eq 3 and Section 3.1: Indeed, there is a typo, the posterior covariance should be $\Sigma^*_{\text{SGPR}}:= K_{ww} - K_{wu}(K_{uu}^{-1} - K_{uu}^{-1} S_u K_{uu}^{-1})K_{uw}$ so that RR’ is describing $K_{uu}^{-1} - K_{uu}^{-1}S_u K_{uu}^{-1}$; we became aware of this when writing the Appendix and corrected it there (please see the note beginning on line 675).
> - Sparse / Stochastic (Gaussian Processes): Indeed, this seems to be a failing mode of the current acronyms for these types of Gaussian processes, we will clarify to define sparse Gaussian processes, what we call SGPR (in the context of Titsias ‘09’s model) as VSGPs to better describe their variational origins and to distinguish them from the model described in Hensman et al, ‘13 which is variationally sparse, but crucially stochastic in nature.
> - Experimental section: We’d strongly encourage you to take a look at the experimental section, which is one of the critical components of the paper and motivates our extensions of O-SGPR. Specifically, our experiments demonstrate that stochastic variational Gaussian processes are very strong benchmark models for both Bayesian optimization and online decision-making, with an additional amount of flexibility that standard Gaussian processes do not have (e.g. ability to handle non-Gaussian likelihoods and a lower memory footprint). Furthermore, our experiments demonstrate that SVGPs can be used in the context of any acquisition function, including lookahead acquisitions and the knowledge gradient. To the best of our knowledge, this is the first time in the literature that SVGPs have even been considered with these acquisitions.
> - Limitations: Indeed, in Appendix A, we do point out a significant amount of limitations of our approach including that the Laplace approximation is likely to fail over time if not updated with real data. We invite the reviewer to take a look at that section for further comments.
>
> **References**
>
> Bui, T. D., C. V. Nguyen, and R. E. Turner. "Streaming sparse Gaussian process approximations." Advances in Neural Information Processing Systems 2017 (2017): 3300-3308.
>
> Bui, Thang D., Josiah Yan, and Richard E. Turner. "A unifying framework for Gaussian process pseudo-point approximations using power expectation propagation." The Journal of Machine Learning Research 18.1 (2017): 3649-3720.
>
> Burt, D., Rasmussen, C. E., and Van Der Wilk, M. (2019).  Rates of convergence for sparse variational Gaussian process regression. In Proceedings of the 36th International Conference on Machine Learning, volume 97, pages 862–871. PMLR.
>
> Jiang, Shali, et al. "Efficient nonmyopic Bayesian optimization via one-shot multi-step trees." Advances in Neural Information Processing Systems 33 (2020).
>
> Osborne, M. A. (2010). Bayesian Gaussian processes for sequential prediction, optimisation and quadrature (Doctoral dissertation, Oxford University, UK).
>
> Rasmussen, Carl, and Williams, Chris. Gaussian Processes for Machine Learning, MIT Press, 2006.
>
> Vakili, Sattar, Victor Picheny, and Artem Artemev. "Scalable Thompson Sampling using Sparse Gaussian Process Models." arXiv preprint arXiv:2006.05356 (2020).

---

> > ### Comment · Reviewer_DFYr · 2021-08-15
> > **Response to rebuttal**
> >
> >
> > I thank the reviewers for the clarifications
> >
> >
> > As a general comment, I like the idea of having fast posterior updates for sparse GPs to easily implement look ahead acquisition functions for sequential decision making, and the experiments demonstrate that the method does the intended job. In the chosen variational framework, the authors indeed need a parameterization, a way to update it and a way to select inducing points.
> >
> > I have read the results section in details and have comments regarding those. In this thread my comments remain on the remainder of the manuscript.
> > As the manuscript stands and with the authors reply, I still have questions and worries about each of these ingredients of the method.
> >
> >
> > ## Regarding the parameterization and the online inference
> >
> >
> > The paper is proposing a particular parameterization of the posterior via sites, i.e. the posterior q(u) is the product of a prior p(u) and Gaussian sites t(u) parameterized via the variables {C,c}. The parameters c,C correspond to natural parameters in exponential family up to a projection. In that sense, the parameterization is well established and not new.
> >
> > In that light it is no surprise that 'adding new data' with a Gaussian observation model amount to linear (additive) updates of these natural parameters. This is well established in the EP literature, including for GPs (see the other Bui reference, eq 18).
> >
> > I believe this deserves a thorough review and discussion. It is more than Woodbury in action, which does not provide clarification. The key challenge of Bui et al (streaming) was to learn the hyperparameters, which lead to the need to directly optimize the bound. without this, which is what the authors do. It would be fair to mention that.
> >
> >
> > ## Regarding the Laplace approximation
> >
> >
> > Thanks for correcting the typo. The corrected objective still appears worrying to me.
> > As presented the optimization is performed under a model whose only observations are that contained in the batch, so past information is forgotten.
> > Would it not be more natural to perform the laplace approximation under (or as the authors put it regularized by) the current posterior?
> > Here my comment on performing approximate inference 'in the context of the current approximation' still stands.
> >
> >
> >
> > ## Regarding the inducing point selection
> >
> >
> > I feel this needs a bit more justification / clarification.
> > It is not obvious from what the authors write that the inducing points are chosen from the set of {the onging ones + the inputs x of the current batch}, and why does it make sense?
> > The Burt lemma is probably the trickiest element introduced in the manuscript. Yet it is the shortest section in the manuscript. It deserved more discussion. epsilon is an a priori expected reconstruction error (exact kernel - nystrom approximation) under the regression observation model before seeing the data and it is -up to a scaling- a bound on the KL between the approximating posterior process and the true posterior process. (see Wild et al 2021, sec 4.1)
> > The authors use it here under the observation model for the summarized pseudo data and the new data in the batch.
> > It does make intuitive sense, but how you derive it and what is happening is not clear from the description.
> > The terminology homo/heteroscedasticity is not helpful here, the observation are not connected to the same random variable (u rather than f), which is the most important difference (not discussed).
> >
> >
> >
> > ## Regarding the conditioning and method name.
> >
> >
> > I understand the meaning of conditioning in Bayesian inference. Most papers in the field propose to derive methods to compute approximate posterior distributions, which is a precise conditional distribution: latent given observations through an observation model.  Here we are talking about a particular model with Gaussian Process prior for the latent.
> > OVC is vague. I am happy with online, variational refers to 'approximate inference'.
> > Why add conditioning? And why not mention Gaussian Processes (GP) since it is tailored to such priors?
> >
> >
> >
> > ## Overall opinion
> >
> >
> > My general feeling is that the full method (parameterization, inducing point selection, laplace approximation) is introduced as a bag of heuristics with partial past work review and little intuition or justification.
> > The fact that the methods works in practice does not overshadow the fact that the method is not enough justified, discussed, and situated in the past literature.
> >
> >
> > For these reasons I maintain my assessment that I believe the manuscript needs substantial extra work to improve its clarity and justification of the method and could then be resubmitted for publication.

---

> > > ### Author Response · Authors · 2021-08-17
> > > **Further Clarifications**
> > >
> > > Thanks for the updated comment. We appreciate your response, and are more than willing to engage in constructive dialogue to improve the overall presentation of the method. The concerns you have maintained after our initial response can be addressed before releasing the camera-ready.
> > > There are many possible connections to previous literature that we do not explore in the paper, due to space limitations and a desire for simplicity.
> > >
> > > For example, there is a deep connection between variationally sparse GP models and EP as described in Bui, Yan, and Turner ‘17. This connection is also present between OVC and EP, as you point out. However, the reader cannot appreciate these connections without a detailed description of variational GP models, our approach, and EP, which is more than is necessary to understand the key ideas (and their attributions) behind OVC.
> > >
> > > As we have already noted in our previous response, we will augment our existing literature review with any relevant references you bring to our attention. We will also include further pointers to the work of Bui, Nguyen, & Turner ‘17 which inspired our approach. Finally, we will also improve the explanatory portions of the paper which appear to currently be confusing.
> > >
> > > **Parameterization and online inference**
> > >
> > > Indeed, the c, C parameterization does end up being the natural parameterization for the exponential family. However, we would appreciate it if you could provide a reference for this parameterization within the GP literature because we are unable to find one that defines it as the “natural parameterization for the exponential family”. We’re happy to cite and discuss this parameterization, but would like to know where it comes from.
> > >
> > > The closest usage of this parameterization in the variational GP literature we can find is the older preprint of Panos, Dellaportas, & Titsias, '18; however, they do not discuss the exponential family connection, and neither of the two papers they cite do either (Opper & Archambeau, ‘09 and Damianou, Titsias, and Lawrence, ‘11). We will of course cite these papers as motivation for the c / C natural parameterization.
> > >
> > > Furthermore, bringing in an approach from another side of the scientific literature should only count as a positive contribution in our favor, rather than something that undermines our work. From what we can tell, the parameterization described in Eq. 18 of Bui, Yan, and Turner, ‘17 ends up reducing to something different as we describe here:
> > >
> > > Defining $\tilde T_n: = \sum_{n=1}^N T_{2,n}, $ and $\tilde Q_{ff}:= K_{fu} K_{uu}^{-1} T_{2,u}^{-1} K_{uu}^{-1} K_{uf}$, then Eq. 18 and its counterparts simplify in the following way:
> > > $$
> > > -Q_{ff} + \tilde Q_{ff} = K_{fu}K_{uu}^{-1}(-K_{uu} + T_{2,u}^{-1})K_{uu}^{-1} K_{uf}
> > > $$
> > > We will call the inner part $(a)$ and then simplify as:
> > > $$
> > > (a) := -K_{uu} + T_{2,u}^{-1} = -K_{uu} + (K_{uu}^{-1} + \tilde T_{n})^{-1}
> > > $$
> > > $$
> > >  = -K_{uu} + K_{uu} - K_{uu}(K_{uu} + \tilde T_n)^{-1} K_{uu}
> > > $$
> > > $$
> > > = - K_{uu}(K_{uu} + \tilde T_n)^{-1} K_{uu}
> > > $$
> > > From what we can tell, this ends up being slightly distinct from our $c,C$ parameterization.
> > > Of course, the updates themselves (presumably Eq. 20 and the lines beneath) are actually low rank updates to a matrix of size m. However, they do seem to be substantially different updates, even if motivated similarly.
> > >
> > > **Laplace Approximation**
> > >
> > > Indeed, instead of regularizing against $K_{ww}$, we could have alternatively regularized against the variational GP posterior, with covariance $\Sigma^* = K_{ww} - K_{wu}K_{uu}^{-1}(K_{uu} - S_u)K_{uu}^{-1}K_{uw}.$ Thanks for the suggestion.
> > >
> > > We tested out the quality of the approximation as in Fig. 8 and found that it produces qualitatively similar results, shown at https://ibb.co/sVRgC0n and https://ibb.co/F81pg3P . More specifically, it seems to produce latent functions that are closer to the original GP posterior predictive and thus the rollouts are more confident overall.
> > > We will include this figure in the final version.
> > >
> > > Finally, we’d expect less regularization (thereby producing larger latent $f$ values) via using the GP posterior. We demonstrate that this does occur in a simple one dimensional classification problem, available at https://ibb.co/KLNLggf . Note that the orange line is computed with Newton iteration against the original posterior.
> > >
> > > **Inducing Point Selection**
> > >
> > > Thank you to the pointer to Wild et al, ‘21 [arXiv:2106.01121] which we hadn’t seen before (we highlight that the arxival date was after the NeurIPS deadline). However, it is certainly relevant to our work, and we will cite it.
> > >
> > > We would like to clarify a point about the pivoted cholesky factorization. To justify using the joint distribution of the current inducing points and the new batch of data, we argue that one should consider the model over the current inducing points as an exact GP. On conditioning (e.g. the observation of the new batch of data), the updated model is still an exact GP. Therefore, to maintain constant memory overhead, we need to determine the top `m` points that summarize the `m + n_{test}` data points.
> > >
> > > ***Pivoted cholesky of new points and inducing points***
> > >
> > > In the online learning setting, there are only three possible solutions that use a constant amount of `m` inducing points (e.g. constant memory and time in the total number of data points). Solutions that perform computations on the old data are not constant time with respect to the total number of data points.
> > >
> > > One is a heuristic approach to randomly combine or “re-sample” old inducing points and new data which is what Bui et al, ‘17 do. However, this approach is prone to catastrophic forgetting as demonstrated by the top row of Figure 7, and does tend to under-perform in practice (see Figure 2a) even in the i.i.d online learning setting.
> > >
> > > The second possible approach is to prune the combination of old inducing points and new data into a set of size “m”. A strong (not optimal because the problem itself is non-convex) greedy approach is then to compute a pivoted cholesky factorization of the kernel over both old inducing points and new data. This is ultimately what we do, as we describe in the section below.
> > >
> > > The third possible approach is to use k-means instead of pivoted cholesky decompositions. In practice, this reduces to an online k-means problem with the estimated centroids being the locations of the inducing points, and seems practical even if not theoretically guided. This is the approach used by Qi et al, ‘10 and Csato and Opper ‘02.
> > >
> > > ***Connection with Burt et al, Wild et al, and Titsias***
> > >
> > > As mentioned in the previous section, we need to prune the combination of old inducing points and new data into a set of size `m` down from `m + n_{test}`. As we have a GP model over these inducing points and new data, the natural approach is to use a pivoted Cholesky inducing point selection strategy of Burt et al, ‘20 (which is theoretically very closely related to the Nystrom approximation, as pointed out by Wild et al, ‘21).
> > > However, this exact GP has a heteroscedastic likelihood (even in the Gaussian setting), and we need to extend the analysis of Burt et al and Titsias to accommodate the heteroscedastic likelihood.
> > >
> > > We sketch how the choice of inducing locations works by re-doing the analysis of Titsias, ‘09 in the heteroscedastic likelihood setting, that is, assuming $y \sim \mathcal{N}(f, \Sigma_y).$
> > > After considerable algebra (following the derivations in Eqs 13 -17 of Titsias’ preprint (http://www2.aueb.gr/users/mtitsias/papers/sparseGPv2.pdf) ), we then have a heteroscedastic version of the sparse GP regression ELBO (the homoscedastic setting is Eq A.1):
> > > $$
> > > \mathcal{F}(\theta, Z) := \log p(y | 0, \Sigma + Q_{\vec v \vec v}) - \frac{1}{2}tr\left(\Sigma_{\vec y}^{-1/2}(K_{\vec v \vec v} - K_{\vec v \vec u}K_{\vec u \vec u}^{-1}K_{\vec u \vec v}) \Sigma_{\vec y}^{-1/2}\right)
> > > $$
> > > This ends up producing the t (in the notation of Burt et al, ‘20) as our $\varepsilon$ in line 198 of the submission.
> > > We can then plug in their arguments to make the argument that one should consider the pivoted cholesky of the “current inducing points + new batch of data” as a result.
> > >
> > > **Name**
> > > OVC is a method for efficiently learning from an _Online_ stream of data with a _Variational_ GP by _Conditioning_ the variational posterior on a sequence of data batches. We contrast this with batch/offline exact conditioning, in which a single batch of data conditions a GP prior through analytic identities for conditional Gaussian random variables.
> > >
> > > **References**
> > > Bui, Thang D., Josiah Yan, and Richard E. Turner. "A unifying framework for Gaussian process pseudo-point approximations using power expectation propagation." The Journal of Machine Learning Research 18.1 (2017): 3649-3720.
> > >
> > > Bui, T. D., C. V. Nguyen, and R. E. Turner. "Streaming sparse Gaussian process approximations." Advances in Neural Information Processing Systems 2017 (2017): 3300-3308.
> > >
> > > Burt, David R., Carl Edward Rasmussen, and Mark van der Wilk. "Convergence of Sparse Variational Inference in Gaussian Processes Regression." Journal of Machine Learning Research 21 (2020): 1-63.
> > >
> > > Csató, Lehel, and Manfred Opper. "Sparse on-line Gaussian processes." Neural computation 14.3 (2002): 641-668.
> > >
> > > Damianou, Andreas C., Michalis K. Titsias, and Neil D. Lawrence. "Variational Gaussian process dynamical systems." Proceedings of the 24th International Conference on Neural Information Processing Systems. 2011.
> > >
> > > Opper, Manfred, and Cédric Archambeau. "The variational Gaussian approximation revisited." Neural computation 21.3 (2009): 786-792.
> > >
> > > Panos, Aristeidis, Petros Dellaportas, and Michalis K. Titsias. "Fully scalable gaussian processes using subspace inducing inputs." arXiv preprint arXiv:1807.02537 (2018).
> > >
> > > Qi, Yuan Alan, Ahmed H. Abdel-Gawad, and Thomas P. Minka. "Sparse-posterior Gaussian processes for general likelihoods." Proceedings of the Twenty-Sixth Conference on Uncertainty in Artificial Intelligence. 2010.

---

> > > > ### Comment · Reviewer_DFYr · 2021-08-25
> > > > **Further comments**
> > > >
> > > > With apologies for the delay, here are further comments
> > > >
> > > > I did include post submission reference [Wild et al 2021] in my review in order to help the authors and not a criticism for omission.
> > > >
> > > > ## Parameterization and online inference
> > > >
> > > > My comment was not very clear. I mostly meant that parameterizing the effect of data via 'sites' or 'pseudo data' has been common in the past for GPs.
> > > > These factors summarize the effect of the data separately from the effect of the prior
> > > > Your equation 2 can be written as $S_u^{-1} = K_{uu}^{-1} + K_{uu}^{-1} C K_{uu}^{-1}$ which shows how $C$ is used to compute the posterior precision.
> > > > There are then many algorithms to update sites both in EP and variational inference (see maybe khan et al 2017 in the non sparse setting)
> > > >
> > > > Eq 7-8 would gain to being split into : (1) change inducing point base and update {c,C} (2) include new data. (literally the 2 terms in the sum, in reverse order)
> > > > Step 1) is not lossless, which motivates the pivoting, but highlighting that explicitly would be helpful.
> > > >
> > > >
> > > > ## Laplace Approximation
> > > >
> > > > Thanks for the extra experiments. There are indeed differences depending on what prior is used for laplace and that radically changes the learning in the online setting.
> > > > See ritter et al 2018.
> > > >
> > > >
> > > > ## Inducing Point Selection + Pivoted cholesky of new points and inducing points
> > > >
> > > > Thanks, the justification makes a lot of sense. An enlightening interpretation is that it correspond to minimizing the reconstruction error of the nystrom approximation on the set of input (z, x_batch), but by adding some weight to this sum of errors. The weight make sure you still approximate the process at the inducing points well.
> > > >
> > > >
> > > >
> > > > ## Overall
> > > >
> > > > There are many nice ideas and practical results in the paper and I enjoyed the discussion in this thread.
> > > > However the manuscript is just not ready and has clarity and conceptual issues.
> > > > The reviewing period is not the moment to do conceptual thinking, which hasn't been conducted with enough depth at this point.
> > > > So I still believe the paper requires some major revision to be accepted.

---

### Official Review · Reviewer_EXgr · 2021-08-24

**Rating:** 5
**Confidence:** 4

**Summary:**

The authors propose online variational conditioning (OVC), a procedure for efficiently conditioning Stochastic Variational Gaussian Processes (SVGPs) in an online setting that does not require re-training through the evidence lower bound with the addition of new data. The authors propose to efficiently update a posterior distribution after receiving new data; they rely on the idea of a Sparse Gaussian Process Regression (SGPR) model trained on pseudo-data under a block-diagonal Gaussian likelihood, they also provide an extension to non-Gaussian likelihoods by means of the Laplace approximation. The authors show the application of their method in look-ahead acquisitions for Bayesian Optimisation.

**Limitations And Societal Impact:**

The authors could improve their work by extending a discussion regarding the reliability of the methods, particularly in the context of Disease Incidence where the authors showed different applications. Such a discussion might enhance the usefulness of their work and the potential positive or negative societal impact.

**Main Review:**

$\textbf{Originality:}$
The work is novel in the way the authors associate the work of “Streaming Sparse GPs” (O-SGPR) with the well-known idea of a Sparse Gaussian Process Regression (SGPR) model. They  highlight that an O-SGPR can be seen as the SGPR  being trained on pseudo-data under a block-diagonal Gaussian likelihood. Additionally, the authors provide an extension to non-Gaussian likelihoods by means of the Laplace approximation.

$\textbf{Quality:} $
Section 3.1 presents the main theoretical analysis and quality contribution of the paper. The section of experiments explores various important scenarios to show the advantages of the method. Nonetheless, in sections 2 and 3,  the work presents some mathematical typos and a few variables are not properly introduced. Also, the authors are probably missing a broader discussion about the results obtained. For instance:

-Figure 3(a) shows that Exact GP+NEI presents the better performance than Exact GP+KG. We could intuitively expect that SVGP+NEI also presented a better performance than SVGP+KG. Although, when using SVGP+NEI we obtain the poorest performance; in contrast, the SVGP+KG achieved the best performance. It might be useful to discuss about such effect. What changes when using Exact GP+NEI and SVGP+NEI? Why does it degrade the performance?

On the other hand, Figure 1 presents an scenario that makes me wonder about the quality of the approach:

-The authors claim the following regarding Figure 1c: “the SVGP is still able to update its posterior over the latent volatility in response to new data without “forgetting” old observations.”. Nonetheless, the figure 1c shows that the mean predictions and error bars differ much between the top and bottom figures for the old batch of data. Shouldn’t we expect the same mean and error bars behaviour (in top and bottom figures 1c) for the old batch of data, since the model does not “forget” old observations? This figure is lacking of a proper discussion about it.

$\textbf{Clarity:} $
The paper is not well organised, the writing style seems not to be homogeneous along the sections. Particularly, the way the sections 1, 2 and 3 are written differ much of the writing style of the section of experiments. There are some confusing phrases along the document, some acronyms are multiple time introduced or not even introduced. There are some inadequate mathematical notations in the paper and a possible error in equation (3) (Please see the detailed comments and recommendations  at the end after “Significance”). The work lacks of an appropriate connection between the experiments section and the previous methodology sections. For instance, section 2.3 “Bayesian Optimization and Monte Carlo Acquisitions” seems to be isolated. Equations (3), (4), (7) and (8) are never cited in the text. In the experiments sections would be important to mention the use of the Algorithm 1 presented in the paper. Likewise, it would be worthy to link such experiments sections with sections 2 and 3.

-Possible error in Eq. (3). The authors refer to the work of Titsias “Variational Learning of Inducing Variables in Sparse Gaussian Processes” for Sparse GP regression and express equations (2) and (3).  Although, the covariance in Eq. (3):
$\Sigma_{SGPR} = K_{\mathbf{ww}} – K_{\mathbf{wu}}( K_{\mathbf{uu}}^{-1} – S_{\mathbf{u}} )K_{\mathbf{uw}}$, for the predictive distribution seems not to be correct in comparison to Titsias’ work. Such a covariance appears in Eq. (6) of Titsias’ work as:
$\Sigma_{SGPR} = K_{\mathbf{ww}} – K_{\mathbf{wu}}( K_{\mathbf{uu}}^{-1} – B )K_{\mathbf{uw}}$, where $B = K_{\mathbf{uu}}^{-1}(S_{\mathbf{u}})K_{\mathbf{uu}}^{-1}$, and $S_{\mathbf{u}}$ is the covariance of the Variational posterior (it is called $A$ in Titsias’ paper)

$\textbf{Significance:} $
The experiments carried out in the paper might be useful for other practitioners and researchers in the field, particularly the paper might contain unique experimental contribution like the results presented in Figure 3 associated with non-Gaussian likelihoods.

$\textbf{Some detailed comments and recommendations for the authors:} $

-The acronyms GP, OVC, SVGP, ELBO, qKG and BO were introduced many times.

-The acronym SGPR appears in: “An improved conceptual understanding of O-SGPR [9] as SGPR trained...” without being introduced.

-The following phrase might be better written to improve the intelligibility: “First, the compute and memory consumption of exact GPs grows at least quadratically with the amount of data [24, 60], generally limiting them to BO problems with fewer than 1,000 function evaluations.”

-It is not clear the sentence: “Second, they are limited to applications that are modeled well by a Gaussian likelihood”. What does it mean “applications that are modeled well”?

-There is a use of the word “compute” that is not clear, for instance: “First, the compute and memory consumption of exact GPs grows…”, also in “Stochastic variational Gaussian processes (SVGPs) [32] have constant compute and memory footprints and are applicable to non-Gaussian likelihoods”. Is it “computation”? Or “computational”?

-The following phrase is not clear: “…, but they sacrifice closed form expressions for conditional posteriors.”. Is it missing additional explanation?

-The caption for “Figure 1” is not very informative and a bit confusing. For instance, at the end of the caption appears: “Efficient, closed form condtioning for SVGPs was previously impractical.”, there is a typo in “conditioning” and what is the author referring to when saying “...was previously impractical”? What is the intention of such a phrase in the caption?

-In section 2.1, typo in “Given $X_{train} = [x_i , . . . , x_n ]$”, change “i” for number “1”?

-The following expression in section 2.1 is not clear: “Computations with $K_{\mathbf{vv}}$ cost $O(n^2 )$ space and $O(n^3 )$ computation.”.

-The matrices $K_{\mathbf{ww}}$ and $K_{\mathbf{vv}}$ appear in section 2.1, but they are not clearly introduced.

-The matrices $K_{\mathbf{uu}}$ and $K_{\mathbf{uv}}$ appear in section 2.2, but they are not clearly introduced.

-Typo in section 2.2, it says: “The optimal $m_{\mathbf{u}}$, $S_{\mathbf{u}}$, and resulting $q(\mathbf{w})$ are”, the variable “$m_{\mathbf{u}}$” should be bold, $\mathbf{m_u}$.

-Possible error in Eq. (3). The authors refer to the work of Titsias “Variational Learning of Inducing Variables in Sparse Gaussian Processes” for Sparse GP regression and express equations (2) and (3).  Although, the covariance in Eq. (3):
$\Sigma_{SGPR} = K_{\mathbf{ww}} – K_{\mathbf{wu}}( K_{\mathbf{uu}}^{-1} – S_{\mathbf{u}} )K_{\mathbf{uw}}$, for the predictive distribution seems not to be correct in comparison to Titsias’ work. Such a covariance appears in Eq. (6) of Titsias’ work as:
$\Sigma_{SGPR} = K_{\mathbf{ww}} – K_{\mathbf{wu}}( K_{\mathbf{uu}}^{-1} – B )K_{\mathbf{uw}}$, where $B = K_{\mathbf{uu}}^{-1}(S_{\mathbf{u}})K_{\mathbf{uu}}^{-1}$, and $S_{\mathbf{u}}$ is the covariance of the Variational posterior (it is called $A$ in Titsias’ paper).

-As a recommendation for section 3.1, it would be useful to apply some intuition to the notation. For instance, it is confusing for the reader to associate the definitions:
“inducing function values $\mathbf{u}^{\prime} := [f(\mathbf{z}_1), . . . , f(\mathbf{z}_p )]$” and
“$\mathbf{u} := [f (\mathbf{z}^{\prime}_1), . . . ,f(\mathbf{z}^{\prime}_p )]$”.
It might be easier to associate the “$\mathbf{u}^{\prime}$” with the evaluations of the “$f(\mathbf{z}^{\prime})$” and also the “$\mathbf{u}$” (without prime) with the evaluations of the “$f(\mathbf{z})$” (without prime).

-In section 3.1, the following notations should be corrected to make the dimensionality coincide:
“… Eq. (6) to define a new SGPR model with pseudo-data ($\hat{X} := [X_t \quad Z_{t-1}]^{\top} , \hat{\mathbf{y}} := [\mathbf{y}_t \quad \tilde{\mathbf{y}}_\mathbf{u^{\prime}}]^{\top}$)”.
A possible way is to write:

$\hat{X}:= [{X_t}^{\top}$ $Z^{\top}_{t-1}]^{\top}$ and $\hat{\mathbf{y}} := [\mathbf{y}^{\top}_t \quad \tilde{\mathbf{y}}^{\top}_\mathbf{u^{\prime}}]^{\top}$.


-In section 3.1, the distributions “$q_t(\cdot)$”, “$\hat{p}(\cdot|\mathbf{y}_t)$” and $p_t(\cdot)$ are not explained.

-In section 3.1, there is not explanation of what is the variable “$\mathbf{\hat{v}}$” in the phrase: “where the distribution of $\mathbf{y}$ is the pseudo-likelihood $N(\mathbf{\hat{v}}, Σ_{\hat{\mathbf{y}}} )$”

-In section 3.1, typo word, Should it be “through”? In phrase: “projecting that distribution back into pseudo-data to be combined with a fresh batch of ground-truth data though a joint Gaussian pseudo-likelihood.”.

-Equations (3), (4), (7) and (8) are never cited in the text.

-There is not a clear explanation for the Figure 2. Therefore, it is difficult to understand what are the main comparisons to analyse between the Figure 2 (b), Figure 2 (c) and Figure 2 (d). Additionally, the colour bars of the figures have different ranges making difficult to distinguish their characteristics. For instance, Figure 2 (b) has a colour bar range (1.6,2.5), Figure 2 (c) has a colour bar range (1.7,2.5) and Figure 2 (d) a colour bar range (1.52,2.16).

-In the caption for Figure 3, the authors refer to NEI (an acronym not introduced previously). For instance in the caption: “Preference learning; SVGPs with qKG slightly outperform Laplace approximations with NEI, and outperform qNEI with SVGPs.”. is the acronym NEI the same as qNEI?

-The authors write in the caption of Figure 3: “Only SVGPs can be used here,”. Nonetheless, there is not information of why only SVGP can be used. It might be useful to give some insight to the reader; probably the caption is not the appropriate place to give such an insight.

-How did the authors select the batch sizes q? It might be useful to have an analysis of the influence of the batch size. What could it be the suggestion for a practitioner when selecting the batch size q?


-In experiments for Hotspot Modelling it appears: "We model the responses $y$ (incidence) at locations $x$ with population $n(x)$ with a Binomial likelihood $p(\hat{y}|f, x)$". is it $y$ different to $\hat{y}$?


Check References:

-In [3], [5], [6], [13], [14], [15], [16], [28], [30], [34], [35], [36], [37], [39], [41], [42], [45], [48], [49], [50], [53], [56], [61], [63], [68] and [79] change “gaussian” by “Gaussian”.

-In [8] change “bayes” by “Bayes”.

-In [20], [21], [23], [43], [48], [51], [73] and [74] change “bayesian” by “Bayesian”.

-In [22] change “svm” by “SVM”.

-In [44] change “lipschitz” by “Lipschitz”.

-In [58] change “hessian” by “Hessian”.

-In [72] change “monte carlo” by “Monte Carlo”.

**Time Spent Reviewing:**

5

---

> ### Comment · Area_Chair_qFpk · 2021-08-24
> **Please, provide your comments for this additional review**
>
> Dear authors,
>
> There is a new review that I commissioned for your paper given the different views of other reviewers. Would you please comment on it as soon as possible?
>
> Many thanks,

---

> ### Author Response · Authors · 2021-08-25
> **Thanks for the review (pt 2/2)**
>
> Thank you for the review and detailed feedback. We leave the response to the detailed comments in this response, and the other comments in the other one.
>
> **Detailed Comments**
>
> Thank you for all of the detailed comments and presentation suggestions, which we will correct.
> - “Applications modelled well” That is, we mean applications where the response function to be optimized is either Gaussian, or is at least continuous. There are several examples in the paper of applications that are not well-modelled by a Gaussian likelihood, such as the volatility model or preference learning.
> - “What do you mean by compute?” We use ‘compute’ in a sense similar to FLOPS (like operation count), but choose the more general term which accounts for other factors such as hardware acceleration. We will make the meaning more clear in the text.
> - “[SVGPs] sacrifice closed form expressions for conditional posteriors.” As we say in the second paragraph of the intro, a big reason exact GP regression models are ideal for online decision-making is because they have closed form conditional posteriors that can be efficiently computed thousands of times during the ‘inner loop’. In contrast, SVGPs do not have closed form posteriors, and therefore they are much more challenging to apply to tasks like active learning or black-box optimization.
> - Figure 1 caption: Apologies for the imprecision and typo in the caption. By “previously impractical”, we mean that it was impossible to produce a plot like the bottom row without a lot of re-training of both the variational distribution and the kernel hypers for the two SVGP models.
> - $X_{\text{train}}$: Yes, should be 1.
> - Computations with $K_{\mathbf{vv}}$: By that we mean, $K_{\mathbf{vv}}$ is a $n \times n$ dense matrix, so storage requires keeping around $n^2$ elements and inverting it (or computing $K_{\mathbf{vv}}^{-1} z$) requires $n^3$ operations (thus time).
> - Definitions for $K_{\mathbf{ww}}$, $K_{\mathbf{vv}}$, $K_{\mathbf{uu}}$, $K_{\mathbf{uv}}$: As stated in sections 2.1 and 2.2, v is the function values at the training locations $X_{train}$, $w$ is the function values at the test locations $X_{test}$, and $u$ is the function values at the inducing point locations Z. $K_{ab}$ is always the prior covariance matrix between function evaluations a and b, e.g. $K_{\mathbf{vv}} = k(X_{train}, X_{train}), K_{uv} = k(Z, X_{train})$, etc.
> - Intuition about notation: Another common convention for kernel matrix notation uses the inputs as the subscripts, e.g. $K_{XX} = k(X, X)$, $K_{XZ} = k(X, Z)$, etc. We found that convention to be unwieldy when writing section 3, opting instead for the current notation.
> - “Through” Yes, although “using” might be helpful as well. We will clean up the phrasing of that sentence.
> - Figure 2: Yes, we will set the scales to be the same as well as including the observed data points that we are conditioning on to make the presentation more obvious. In this figure, our goal is to show that the minimizers of the KG acquisition surface across both the exact GP and the SVGP with exact conditioning are very close to each other (around 1.5, -2.0). By comparison, not conditioning into an SVGP, but conditioning into a SGPR model (labelled as O-SGPR conditioning) produces a very different minimizer (around 0.0, -4). The maximizers are located similarly.
> - NEI: Yes, that should be qNEI.
> - “Only SVGPs…” Only SVGPs (and not standard GPs) can be used here due to the non-Gaussianity of the data.
> - Selection of batch sizes q: Good question, in all experiments, we mimicked the batch sizes of each experimental setup that we referenced. For example, we chose a batch size of q = 3 in Figures 3a,b as that’s what is in the tutorial we started from: https://botorch.org/tutorials/closed_loop_botorch_only, but a batch size of $100$ in the rover experiments as that’s what’s done in Wang et al, ‘18. Please see the ablation study for an empirical comparison on the constrained Hartmann problem.
>
>     - Practically, we’d suggest using the largest batch size that can be used while still maintaining computational efficiency in evaluating the expensive true function. For example, in biological laboratories, running several experiments in parallel is straightforward, whereas an expensive physical simulator may only be able to run sequentially.
>
>     - Finally, see results such as Fig 5 in Wu & Frazier, NeurIPS, '16 (https://arxiv.org/abs/1606.04414) for a demonstration of qKG across various levels of q and many of the figures in Wilson et al, NeurIPS, '18 (https://arxiv.org/abs/1805.10196) for various acquisitions across several batch sizes and test problems. Note that this question is closely related to the so-called “adaptivity gap” in the active search literature (Jiang et al, NeurIPS, '18, https://papers.nips.cc/paper/2018/file/a7aeed74714116f3b292a982238f83d2-Paper.pdf) and deserves further empirical study on a problem specific basis.
>
> - Hotspot modelling: Yes, that is a typo - should be $y$.
>
> Thank you for the pointers in the references, we will correct these.

---

> ### Author Response · Authors · 2021-08-25
> **Thanks for the review (pt 1/2)**
>
> Thank you for the review and detailed feedback. Here is the first part of the response, focusing primarily on the broader implications, quality, and clarity portions of the review.
>
> **Broader Impact and Implications**
>
> We’d like to emphasize that our experiments demonstrate that SVGPs are extremely useful as a base model for lookahead Bayesian optimization. This is a sea change in practice, as it enables practitioners to use advanced acquisition functions in problems where there are non-Gaussian observations (e.g. Poisson, Binomial, etc.). Using SVGPs, as we demonstrate, additionally gives computational improvements when using BO to solve complicated control based problems where large numbers of function evaluations are needed. As we mention in Appendix A, both of these issues are closely related to real-world problems (think of the public health examples in the paper).
>
> **Ablation on Noise Level and Batch Size**
>
> We provide a couple of ablation studies on the Hartmann6 test problem that we hope explains some of the results in Figure 3(a) as well as parsing out the importance of the batch parameter mentioned in the detailed comments portion of the review. Recall that we consider a noise standard error of 0.5 and q=3 for this problem in the main text of the paper.
>
> As an ablation study, we ran 20 trials with a low noise setting (se = 0.1) or high noise setting (se = 0.5) and varied the batch size (q = 1 or q = 3), finding that the mean maximum end of trial achieved for the methods was as follows:
>
> In the first set of experiments, we considered a batch size of 1:
>
> | Noise Level     | Acqf     | Method     | Best Value       |
> |-------------    |------    |--------    |--------------    |
> | 0.1             | NEI      | Exact      | 3.14 (0.13)      |
> | 0.1             | NEI      | SVGP       | 3.10 (0.095)     |
> | 0.1             | KG       | Exact      | 3.20 (0.02)      |
> | 0.1             | KG       | SVGP       | 3.18 (0.03)      |
> | 0.5             | NEI      | Exact      | 3.10 (0.10)      |
> | 0.5             | NEI      | SVGP       | 3.10 (0.14)      |
> | 0.5             | KG       | Exact      | 3.12 (0.04)      |
> | 0.5             | KG       | SVGP       | 3.16 (0.05)      |
>
> In the second set of experiments, we considered a batch size of 3 (as in the paper):
>
> | Noise Level     | Acqf     | Method     | Best Value      |
> |-------------    |------    |--------    |-------------    |
> | 0.1             | NEI      | Exact      | 3.19 (0.05)     |
> | 0.1             | NEI      | SVGP       | 3.08 (0.06)     |
> | 0.1             | KG       | Exact      | 3.17 (0.03)     |
> | 0.1             | KG       | SVGP       | 3.13 (0.04)     |
> | 0.5             | NEI      | Exact      | 3.20 (0.03)     |
> | 0.5             | NEI      | SVGP       | 3.12 (0.06)     |
> | 0.5             | KG       | Exact      | 3.15 (0.03)     |
> | 0.5             | KG       | SVGP       | 3.21 (0.05)     |
>
> In general, we find that on this problem, the q=3 choice is slightly better than q=1 (but they are extremely comparable). Similarly, we see that SVGP + KG tends to perform better at higher noise levels than both Exact + NEI and Exact + KG. We attribute this to slightly better noise robustness in the model fitting stage (after all, the SVGP has considerably more free parameters to adapt). We will include this ablation in the camera ready.
>
> Note that we used the first 20 results for the q=3, high noise setting we had previously run (not 50 trials like in the paper).
>
> **Quality**
>
> - Figure 3(a): Indeed, this was a bit surprising to us as well. As we added Gaussian noise into the function, we might a priori expect that qNEI should outperform qKG given the class of models. However, all things being held equal, a model that is more robust to the observed noise should tend to perform better, particularly if we are using more than just its mean and variance. This seems to be what’s going on comparing the SVGP and exact GP fits. See the ablation study for further details.
> - Figure 1(c): Yes, the prediction does change a bit in the farthest right two panels. On taking a detailed look at the experimental setup, we realized that the latent mean prediction does not change, but rather the differences can be attributed to two factors:
>
>     _a)_ the non-linear transformation and our usage of 10 random samples from the posterior over `f` to compute the mean and standard deviation in volatility space
>
>     _b)_ the non-convexity of the volatility likelihood which yields a Hessian w.r.t f at each data point that is ill-conditioned and dependent on the pseudo-response \hat f. The second issue necessitates a lot of numerical jitter being added into the predictive mean computation (which could produce some changes in latent space), while the first introduces randomness directly in response (volatility) space.  We will include a plot of the latent mean prediction (which does not change) in the camera ready.
>
> **Clarity**
>
> Thanks for the pointers, we will incorporate them into the camera ready.
>
> Eq 3: Yes, that is indeed a typo, which we will correct. We became aware of this typo when writing the Appendix and corrected it there, see lines 675-681.
>
> **Reliability**
>
> Please see Appendix A, where reliability and societal impacts are described in some detail. As we describe in lines 606-608, over-reliance on machine learning models can potentially lead to over-confidence in their results. In a public health setting, this could be problematic if the model is too overconfident that there is not disease in an area, thus letting an outbreak spread unchecked.
>
> **Conclusion**
>
> We note that your primary concerns appear to be about style and presentation. We hope our responses have helped clarify. We will indeed make every effort to incorporate your feedback, which we really appreciate. We also note that several reviewers do highlight the quality of presentation as a general strength of the paper. We believe these points can be straightforwardly addressed in a camera ready version.

---

### Author Response · Authors · 2021-08-10
**To All Reviewers**

We thank the reviewers for their thoughtful feedback. Our paper addresses the timely and important problem of scalable online updates for probabilistic modeling. While there has been a flurry of recent activity making progress in scalable Gaussian processes, advances in scalable online inference have been relatively scarce, despite their high significance.

In particular, we propose OVC (online variational conditioning) as a solution to the conditional updates problem for Gaussian processes (i.e. fantasization). That is, given a variational Gaussian process model that has a predictive posterior distribution p(f | D), where D is the data, how do we update the posterior distribution to compute p(f | D, D’) where D’ is newly observed data? As we demonstrate throughout the experiments of the paper, this foundational problem is often at the heart of online probabilistic modelling and decision-making. As all of the reviewers pointed out, our approach to tackling this problem is both novel and timely.

Our work re-interprets and builds off of the work of Bui et al, ‘17 who proposed O-SGPR, which implicitly updates the variational distribution to update a sparse variational Gaussian process model in response to new data. However, O-SGPR itself is tied to Gaussian likelihoods for the responses. Our extensions to O-SGPR, namely a mechanism for updating inducing point locations and the use of Laplace approximations for non-Gaussian observations, enable us to make SVGPs a strong building block for Bayesian optimization and online learning tasks. We believe that our paper i) specifically demonstrates the power of SVGPs in Bayesian optimization, ii) resolves the challenges of using SVGPs inside of online learning tasks, and iii) moves towards the unification of model conditioning for both exact Gaussian processes and SVGPs.

Specifically, in these tasks, we loop the following two steps: first, we fit a SVGP (both variational distribution and kernel hyperparameters) on the observed dataset, and second, we use the fitted SVGP to acquire new data points by optimizing an acquisition function (either with gradient based approaches as in our online learning or Bayesian optimization experiments or via Thompson sampling as in our control experiments).

We’d also like to emphasize that our contribution goes beyond the methodological development of OVC. Our experiments demonstrate that using OVC makes using SVGPs practical and high performing in a wide variety of Bayesian optimization tasks. For example, we develop rollout based approaches to Thompson sampling that are much more practical with SVGPs due to the enhanced numerical stability over exact GPs. We then use these strategies to achieve very strong performance on several control problems (see Section 4.3) including on several MuJoCo tasks (swimmer and hopper) where Bayesian optimization approaches have previously been dominated by both control and reinforcement learning based approaches. We additionally demonstrate the feasibility of optimizing latent functions after only having observed non-Gaussian observations with SVGPs, significantly reducing the user overhead for these classes of problems which range from preference learning (Fig. 3.d) to disease tracking (Fig 4b,c).

We respond individually to each reviewer in separate posts. We hope the reviewers will consider our responses, in addition to the timeliness and significance of the contribution, in their final assessment.

---

### Public Comment · ~Thang_D._Bui1 · 2022-04-23
**interesting work but the similarities + differences to Bui, Nguyen and Turner (2017) could be made more explicit**

Dear authors,

Thanks for the interesting paper that demonstrates the utility of fast and accurate online inference for sequential decision-making tasks.

After reading the OVC paper and the reviews/discussion here, I would like to clarify the following points in relation to our streaming sparse Gaussian process approximations (SSGP) paper:

+ SSGP is not tied to Gaussian likelihoods. The proposed variational bound is more general and was, in fact, used for a toy classification example in the paper.
+ SSGP allows for online optimisation of hyperparameters which your paper does not consider as hypers were kept fixed [this question was raised by a few reviewers]. The OVC paper mentioned the SSGP presentation is "technical", but it is technical exactly because of this hyperparameter issue.
+ the closed-form optimal variational posterior and the resulting collapsed bound for SSGP regression were explicitly discussed in sec 3.2 of our paper. The "pseudo-data" or "effective-likelihood" view and the similarity to SGPR are fairly clear from these equations.

All these comments aside, I agree with the pitfalls of SSGP as discussed in this paper (e.g. small online batch + hard to re-optimise inducing locations which are important for tasks like BO), and that the proposed method in the paper fixes these issues.

Thanks,
Thang

---

> ### Public Comment · Authors · 2022-07-20
> **Thanks for your comment**
>
> Hi Thang,
>
> Thanks for reading and engaging with us and apologies for the long delay in replying. We were deeply influenced by your work and hope that the paper reads accordingly. Our work originally began with a deep investigation of SSGP (some early experiments are actually shown in the Appendix of https://arxiv.org/abs/2103.01454). As you are well aware this material is very technical, so if you have any further specific concerns we would be happy to discuss them.
>
> - **non-Gaussian likelihoods:** the variant of SSGP we refer to as O-SGPR in our paper is specifically the collapsed bound variant presented in Eqs. 6-8 in https://arxiv.org/pdf/1705.07131.pdf, which is tied to a Gaussian likelihood. We agree the uncollapsed bound presented in Eqs. 13-14 is more general, and regret any confusion we may have caused on this point.
>
> - **optimizing hyperparameters online:** we explicitly consider hyperparameters varying in time in Section 3.1 of our paper (specifically equations 6 - 8), and so OVC does in fact allow hyperparameters to be learned online through the fantasization procedure.
>
> - **novelty of the pseudo-data view:** The pseudo-data view as discussed in our paper was the result of repeated careful readings of your paper (especially Section 3) and considerable independent thought trying to digest your paper and its presentation. Please see appendix C.2 and the end of Section 3.1 for an extended discussion on the relationship between the pseudo-data view of OVC, the collapsed bound SSGP variant, and SGPR, which gives an intuitive interpretation of the pseudo-data that we did not find explicitly discussed in your paper.
>
> Best,
>
> the authors

---

### Decision · Program_Chairs · 2021-09-27

**Decision:**

Accept (Poster)

**Comment:**

The paper contributes to the literature of sparse variational Gaussian processes applied to an online setting. The authors introduce the novel idea of treating inducing variables and inducing inputs at time t-1 as pseudo-data to be used at time t, leading to recursive expressions for the posterior distribution. The paper appears technically solid. The most critical reviewer does not oppose acceptance. Comments regarding clarity and typos highlighted by the reviewers can be addressed in the camera-ready version.